# Neurocalcin regulates nighttime sleep and arousal in *Drosophila*

Ko-Fan Chen*, Simon Lowe, Angélique Lamaze, Patrick Krätschmer, James Jepson*

Department of Clinical and Experimental Epilepsy, UCL Institute of Neurology, London, United Kingdom

**Abstract** Sleep-like states in diverse organisms can be separated into distinct stages, each with a characteristic arousal threshold. However, the molecular pathways underlying different sleep stages remain unclear. The fruit fly, *Drosophila melanogaster*, exhibits consolidated sleep during both day and night, with night sleep associated with higher arousal thresholds compared to day sleep. Here we identify a role for the neuronal calcium sensor protein Neurocalcin (NCA) in promoting sleep during the night but not the day by suppressing nocturnal arousal and hyperactivity. We show that both circadian and light-sensing pathways define the temporal window in which NCA promotes sleep. Furthermore, we find that NCA promotes sleep by suppressing synaptic release from a dispersed wake-promoting neural network and demonstrate that the mushroom bodies, a sleep-regulatory center, are a module within this network. Our results advance the understanding of how sleep stages are genetically defined.
DOI: https://doi.org/10.7554/eLife.38114.001

*For correspondence:
kofan.chen@gmail.com (K-FC);
j.jepson@ucl.ac.uk (JJ)

Competing interests: The authors declare that no competing interests exist.

## Introduction

Sleep is a widely conserved behavior that influences numerous aspects of brain function, including neuronal development (*Kayser et al., 2014*), clearance of metabolic waste (*Xie et al., 2013*), synaptic plasticity (*Havekes et al., 2016*; *Kuhn et al., 2016*; *Li et al., 2017*; *Yang et al., 2014*), and complex behaviors (*Kayser et al., 2015*; *Kayser et al., 2014*). The fruit fly, *Drosophila*, exhibits a sleep-like state characterized by immobility, altered posture and elevated arousal threshold during both day and night (*Hendricks et al., 2000*; *Shaw et al., 2000*). Similarly to mammals, sleep in *Drosophila* is regulated by circadian and homeostatic processes (*Huber et al., 2004*; *Liu et al., 2014*). Furthermore, just as human sleep can be separated into stages of differing arousal thresholds (REM and three non-REM sleep stages) (*Rechtschaffen et al., 1966*), sleep in *Drosophila* also varies in intensity throughout the day/night cycle, with night sleep having a higher arousal threshold relative to day sleep (*Faville et al., 2015*; *van Alphen et al., 2013*).

The molecular mechanisms by which sleep is partitioned into stages remain poorly understood. In *Drosophila*, mutations in a select number of genes modulate either day or night sleep, suggesting that distinct genetic pathways may promote or inhibit these sleep stages (*Ishimoto et al., 2012*; *Tomita et al., 2015*). Yet it is still unclear which properties of sleep/wake these genes are influencing, and how the timing of when they affect sleep is controlled.

The identification of new genes selectively impacting day or night sleep will help address such questions. Previously, large-scale screens of EMS-mutagenized (*Cirelli et al., 2005*; *Stavropoulos and Young, 2011*), P-element insertion (*Koh et al., 2008*), or transgenic RNAi knockdown lines (*Rogulja and Young, 2012*) have been used to identify *Drosophila* sleep mutants. However, such approaches are highly laborious, requiring screening of thousands of fly lines to identify a limited number of bona fide sleep genes. Thus, targeted screening strategies of higher efficiency may represent a useful complement to unbiased high-throughput, yet low yield, methodologies.

We uncovered a novel sleep-relevant gene in *Drosophila* using a guilt-by-association strategy. Our approach was based on comparative phenotyping of human and *Drosophila* mutants of homologous genes, *KCTD17*/*insomniac*, both of which encode a Cullin-3 adaptor protein involved in the ubiquination pathway (*Mencacci et al., 2015*; *Pfeiffenberger and Allada, 2012*; *Stavropoulos and Young, 2011*). In humans, a *KCTD17* mutation has been associated with myoclonus dystonia, a disorder characterized by repetitive movements, contorted postures and non-epileptic myoclonic jerks in the upper body (*Mencacci et al., 2015*). In *Drosophila*, null or hypomorphic mutations in the *KCTD17* homolog *insomniac* result in profound reductions in sleep (*Pfeiffenberger and Allada, 2012*; *Stavropoulos and Young, 2011*).

Genotype-to-phenotype relationships arising from conserved cellular pathways can differ substantially between divergent species such as *Drosophila* and humans (*Lehner, 2013*; *McGary et al., 2010*; *Wangler et al., 2017*). In this context, it is interesting to note that dystonia in humans and sleep in *Drosophila* are linked by a common cellular mechanism: synaptic downscaling. This process occurs during sleep in both mammals and *Drosophila*, and is suppressed at cortico-striatal synapses in murine dystonia models (*Bushey et al., 2011*; *Calabresi et al., 2016*; *Gilestro et al., 2009*; *Martella et al., 2009*; *Tononi and Cirelli, 2014*). Thus, we hypothesized that homologs of other human dystonia-associated genes might also influence sleep in *Drosophila*.

To test this hypothesis, we examined whether homologs of dystonia-associated genes influenced sleep in *Drosophila*. Through this strategy we identified a previously unappreciated role for the *HPCA*/Hippocalcin homolog *Neurocalcin* (*Nca*) in regulating night sleep. Hippocalcin and NCA are neuronal calcium sensors, cytoplasmic proteins that bind calcium via EF hand domains and translocate to lipid membranes via a calcium-dependent myristoylation switch. This in turn alters interactions with membrane-bound proteins such as ion channels and receptors (*Braunewell et al., 2009*; *Burgoyne and Haynes, 2012*). In murine hippocampal neurons, Hippocalcin facilitates the slow afterhyperpolarisation (a calcium-dependent potassium current) (*Tzingounis et al., 2007*), and glutamate receptor endocytosis during LTD (*Jo et al., 2010*; *Palmer et al., 2005*). In humans, rare missense and null mutations in *HPCA* have been linked to DYT2 primary isolated dystonia, a hyperkinetic movement disorder affecting the upper limbs, cervical and cranial regions (*Atasu et al., 2018*; *Carecchio et al., 2017*; *Charlesworth et al., 2015*). *Drosophila* NCA has been shown to be expressed in synaptic regions throughout the fly brain (*Teng et al., 1994*). However, the neuronal and organismal functions of NCA have remained elusive. Here, we demonstrate a role for NCA in suppressing nocturnal arousal and locomotor activity in *Drosophila*, thus facilitating nighttime sleep.

## Results

### Identification of neurocalcin as a sleep-promoting factor

*Drosophila* NCA is highly homologous to the mammalian neuronal calcium sensor Hippocalcin, sharing >90% amino-acid identity (*Figure 1—figure supplement 1*). To test whether *Nca* influences sleep or wakefulness we initially used transgenic RNAi. Using the pan-neuronal driver *elav*-Gal4, we found that neuronal expression of three independent RNAi lines targeting *Nca* mRNA (*kk108825*, *hmj21533* and *jf03398*; termed *kk*, *hmj* and *jf* respectively) reduced night sleep but not day sleep in adult male flies housed under 12 hr light: 12 hr dark conditions (12L: 12D) at 25°C (*Figure 1—figure supplement 2A–E*), as measured by the *Drosophila* Activity Monitoring (DAM) system (*Pfeiffenberger et al., 2010*). In this work we define a *Drosophila* sleep bout as ≥5 min of inactivity, the common standard in the field (*Pfeiffenberger et al., 2010*).

We performed a series of experiments to further validate a specific role of NCA in promoting night sleep. Sleep loss in flies expressing *Nca* RNAi correlated with significant reductions in *Nca* expression (*Figure 1—figure supplement 2F*). In contrast, expression of the *cg7646* locus, which shares 5' regulatory elements with *Nca* and encodes a neuronal calcium sensor more closely related to mammalian Recoverin than Hippocalcin, was unaffected by *Nca* knockdown (*Figure 1—figure supplement 2A,G*). Night-specific sleep loss following *Nca* knockdown was also observed in virgin adult female flies and in male flies expressing the *kk Nca* RNAi using other pan-neuronal or broadly expressed drivers (*Figure 1—figure supplement 2H–J*), whereas knockdown of *cg7646* by RNAi did not impact night sleep (*Figure 1—figure supplement 2K*).

Sleep architecture in *Drosophila* is generally studied in 12L: 12D conditions. Interestingly, we found that night sleep in *Nca* knockdown males appeared even further reduced under short photo-period conditions (8L: 16D) (*Figure 1A*). Similarly to 12L: 12D, in 8L: 16D day sleep was unaffected whilst night sleep was reduced (*Figure 1A–C*), due to fragmentation of consolidated sleep bouts during the middle of the night (*Figure 1—figure supplement 3*). Nocturnal sleep loss in 8L: 16D was again observed in flies expressing the independent *hmj* and *jf Nca* RNAi lines in neurons (*Figure 1—figure supplement 4A–C*), but not in flies expressing the *kk Nca* RNAi line in muscle cells (*Figure 1—figure supplement 4D–F*), supporting a role for NCA in neurons.

Given the limited spatial resolution of the DAM system, which measures activity via a single infra-red beam, we undertook a higher resolution analysis of sleep using a video-tracking method - the DART (*Drosophila* ARousal Tracking) system (*Faville et al., 2015*). DART recordings confirmed night-specific sleep loss in in *Nca* knockdown flies housed under 8L: 16D (*Figure 1D–F*).

To test whether sleep loss caused by neuronal *Nca* knockdown flies was due to an indirect effect on the circadian clock, we examined whether *Nca* knockdown altered circadian patterns of locomotor activity in constant dark (DD) conditions. Importantly, knockdown of *Nca* in neurons did not alter circadian rhythmicity (*Figure 1—figure supplement 5*). Furthermore, knockdown of *Nca* specifically in clock neurons did not affect night sleep (see below). Thus, it is unlikely that sleep loss in *Nca* knockdown flies is due to circadian clock dysfunction.

To provide further genetic evidence that NCA is a sleep-regulatory factor, we generated three independent *Nca* null alleles by replacing the entire *Nca* locus (including 5' and 3' UTRs) with a mini-*white*$^+$ sequence using ends-out homologous recombination (*Baena-Lopez et al., 2013*). The mini-*white*$^+$ is flanked by loxP sites, allowing removal by Cre recombinase and leaving single attP and loxP sites in place of the *Nca* locus (*Figure 1—figure supplement 6A–C*). As expected, no *Nca* mRNA expression was detected in homozygotes for the deleted *Nca* locus (*Figure 1—figure supplement 6D–E*). Thus, we term these alleles *Nca*$^{KO1-3}$ (*Nca* knockouts 1–3). Following outcrossing into an isogenic *iso31* control background, male homozygotes and transheterozygotes for the three *Nca* knockout alleles were viable to the adult stage and exhibited normal day sleep but reduced night sleep, as measured by both DAM and DART systems (*Figure 1G–L*, *Figure 1—figure supplement 7*), similarly to *Nca* knockdown flies.

By examining locomotor patterns in individual flies using the DART system, we found that *Nca*$^{KO1}$ males consistently displayed prolonged activity relative to controls following lights-off and frequent bouts of movement even in the middle of the night – a period of quiescence in *iso31* controls (*Figure 1M,N*). Video-based analysis of waking locomotor velocities revealed that loss of NCA led to a reduction in average locomotor velocity across 24 hr (*Figure 1—figure supplement 8A,B*). This was primarily driven by reduced locomotor activity during the evening activity peak and following lights-off, suggesting loss of NCA mildly reduces peak levels of activity (*Figure 1—figure supplement 8C*). In contrast, locomotor velocities during normally quiescent periods of the night were greatly enhanced in *Nca*$^{KO1}$ males compared to *iso31* controls (*Figure 1—figure supplement 8D*), consistent with a perturbed sleep state.

Collectively, the above data demonstrate that NCA promotes night sleep in *Drosophila* and does so by acting in neurons. For simplicity, we use the *kk Nca* RNAi line and the *Nca*$^{KO1}$ knockout line for all subsequent experiments, and refer to these flies as *Nca*$^{KD}$ (*Nca* knockdown) and *Nca*$^{KO}$ (*Nca* knockout) respectively.

## NCA suppresses nighttime arousal

Sleep is characterized by a reduced responsiveness to stimuli (*Campbell and Tobler, 1984*). Recent studies have shown that responsiveness during sleep stages in *Drosophila* is dynamically regulated, with night sleep exhibiting a higher arousal threshold relative to day sleep (*Faville et al., 2015*; *van Alphen et al., 2013*). Since knockout or knockdown of NCA specifically impacted night sleep, we were interested to test whether NCA might also influence the arousal threshold during the night. To do so, we used the DART system to subject *Nca*$^{KD}$ flies and respective controls to a mechanical stimulus consisting of five consecutive 50 Hz vibrations of 200 ms duration, each separated by 800 ms, at either Zeitgeber Time (ZT) 4 (the middle of the day) or ZT16 (the middle of the night) in 8L: 16D (see Methods). This paradigm has previously been shown to induce startle responses the majority of *white* mutant flies sleeping during the day, and a correspondingly smaller proportion when applied during the night (*Faville et al., 2015*).

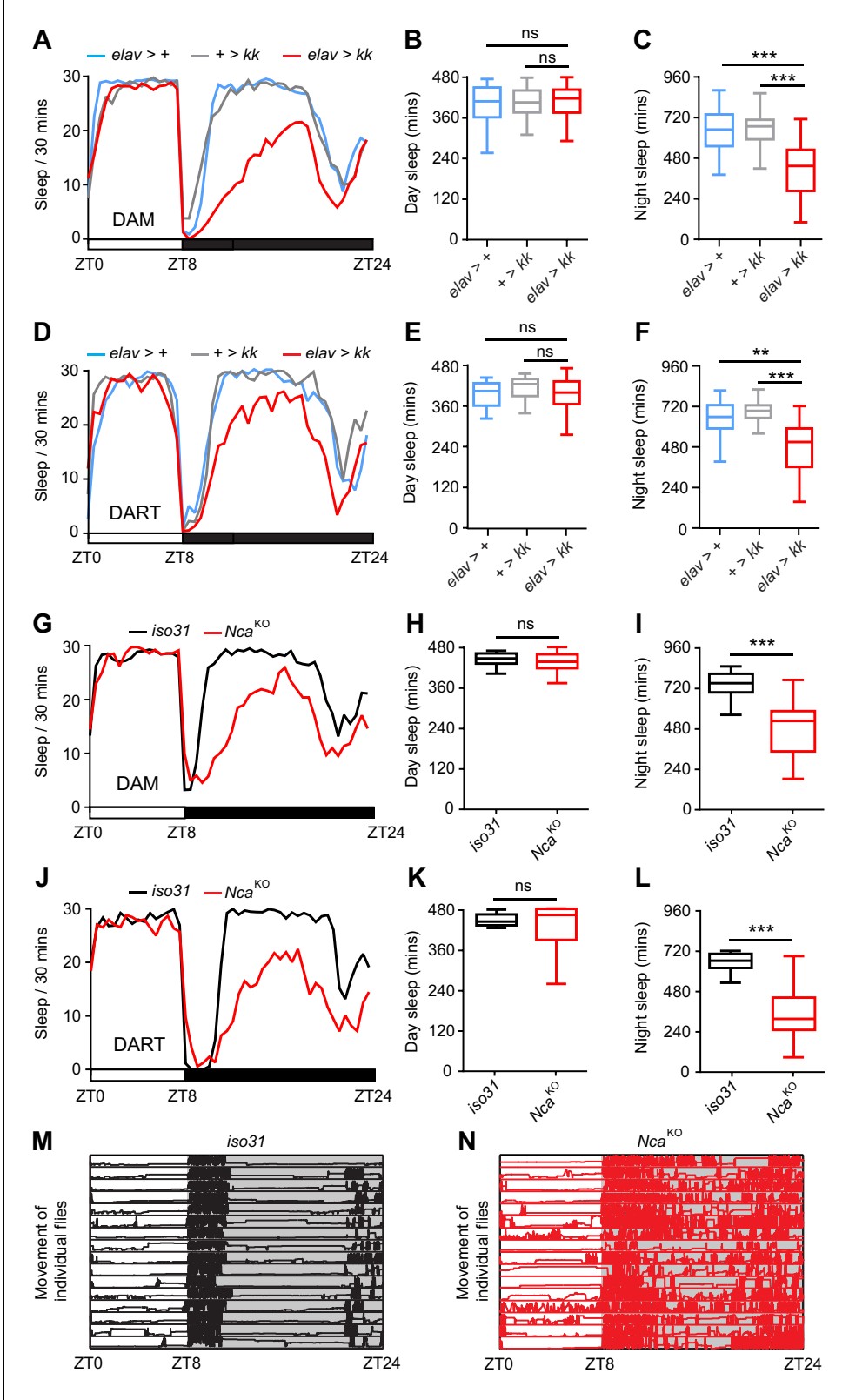

**Figure 1.** Neurocalcin promotes night sleep. (A) Mean sleep levels measured using the DAM system under 8L: 16D conditions for adult male pan-neuronal *Nca* knockdown flies (*elav > kk*) and associated controls (*elav*-Gal4 driver or *kk* RNAi transgene heterozygotes). (B–C) Median day (B) and night (C) sleep levels in the above genotypes. n = 54–55. Data are presented as Tukey box plots. The 25th, Median, and 75th percentiles are shown. Whiskers represent 1.5 x the interquartile range. Identical representations are used in all subsequent box plots. (D) Mean sleep levels measured using the DART

*Figure 1 continued on next page*

*Figure 1 continued*

system in 8L: 16D conditions for male adult pan-neuronal *Nca* knockdown flies (*elav > kk*) and associated controls. (E–F) Median day (E) and night (F) sleep levels in the above genotypes. n = 20 per genotype. (G) Mean sleep levels in 8L: 16D conditions for *Nca*KO adult males and *iso31* controls measured using the DAM. (H–I) Median day (H) and night (I) sleep levels in the above genotypes. n = 32 per genotype. (J) Mean sleep levels in 8L: 16D conditions for *Nca*KO adult males and *iso31* controls measured by DART. (K–L) Median day (K) and night (L) sleep levels in the above genotypes. n = 16 per genotype. (M–N) The longitudinal movement for individual *iso31* (M) and *Nca*KO (N) flies are shown as rows of traces plotting vertical position (Y-axis) over 24 hr (X-axis) under 8L: 16D condition. ns (not significant) - p>0.05, **p<0.01, ***p<0.001, Kruskal-Wallis test with Dunn's post-hoc test (B–C, E–F) or Mann-Whitney U-test (H–I, K–L).

DOI: https://doi.org/10.7554/eLife.38114.002

The following source data and figure supplements are available for figure 1:

**Source data 1.** Sleep, velocity, rhythmicity data and gene expression data from *Nca* knockdown and knockout flies relating to *Figure 1* and associated figure supplements.

DOI: https://doi.org/10.7554/eLife.38114.011

**Figure supplement 1.** Human Hippocalcin and *Drosophila* Neurocalcin are highly homologous neuronal calcium sensors.

DOI: https://doi.org/10.7554/eLife.38114.003

**Figure supplement 2.** Pan-neuronal knockdown of *Nca* using independent RNAi lines causes night sleep loss.

DOI: https://doi.org/10.7554/eLife.38114.004

**Figure supplement 3.** Reduced consolidated sleep in *Nca*KD flies.

DOI: https://doi.org/10.7554/eLife.38114.005

**Figure supplement 4.** Pan-neuronal expression of independent *Nca* RNAi lines results in night sleep loss in 8L: 16D conditions.

DOI: https://doi.org/10.7554/eLife.38114.006

**Figure supplement 5.** *Nca* knockdown does not alter circadian rhythmicity.

DOI: https://doi.org/10.7554/eLife.38114.007

**Figure supplement 6.** Generation of *Nca* null alleles using ends-out homologous recombination.

DOI: https://doi.org/10.7554/eLife.38114.008

**Figure supplement 7.** Independent combinations of *Nca* knockout alleles exhibit night sleep loss.

DOI: https://doi.org/10.7554/eLife.38114.009

**Figure supplement 8.** Locomotor velocities in *Nca* knockout flies.

DOI: https://doi.org/10.7554/eLife.38114.010

At ZT4, we found that the majority of adult males from both control lines exhibited startle responses in response to vibration stimuli, and that *Nca* knockdown did not significantly alter the arousal threshold at this time point (*Figure 2A,B*). In contrast, the percentage of *Nca*KD flies responding to vibration stimulus was significantly higher at ZT16 relative to both control lines (*Figure 2C,D*). Furthermore, whereas the percentage of control flies responding to vibration stimulus was significantly higher during the day compared to the night (*elav > +* and *+ > kk*: p<0.0005, Binomial test with Bonferonni correction for multiple comparisons), there was no significant day/night difference in responsiveness in *Nca*KD flies (p=0.1). Similar results were also observed in *Nca*KO flies (*Figure 2E,F*). These data suggest that NCA is a molecular regulator of nighttime arousal in *Drosophila*.

## Light-sensing and circadian pathways define when NCA promotes sleep

The night-specificity of sleep loss and heightened arousal in *Nca*KO and *Nca*KD flies prompted us to test whether circadian and/or light-sensing pathways determine when NCA impacts sleep. Initially, we examined sleep patterns in *Nca*KD flies under DD conditions, in which the circadian clock alone distinguishes subjective day from night. Interestingly, in DD robust sleep loss in *Nca*KD flies was still restricted to the subjective night (*Figure 3A,B*). These data suggest that the circadian clock intersects with NCA and demonstrate that sleep loss in *Nca*KD flies is not simply due to darkness-induced hyperactivity.

To confirm that the circadian clock influences NCA's sleep-promoting role, we analyzed sleep in *Nca*KD flies under arrhythmic conditions. In a *timeless* knockout (*tim*KO) background in DD (*Lamaze et al., 2017*), where the clock no longer demarcates subjective day from night, sleep loss in *Nca*KD flies was observed throughout the 24 hr dark period (*Figure 3C,D*). Intriguingly, in a *tim*KO background in 8L: 16D conditions, significant sleep loss in *Nca*KD flies was observed during the night (*Figure 3E,F*), but not during the day (*elav > kk, tim*KO vs *elav > +, tim*KO: p=0.75, Kruskal-Wallis test with Dunn's post-hoc test). Thus, light is also capable of inhibiting sleep loss in *Nca*KD flies.

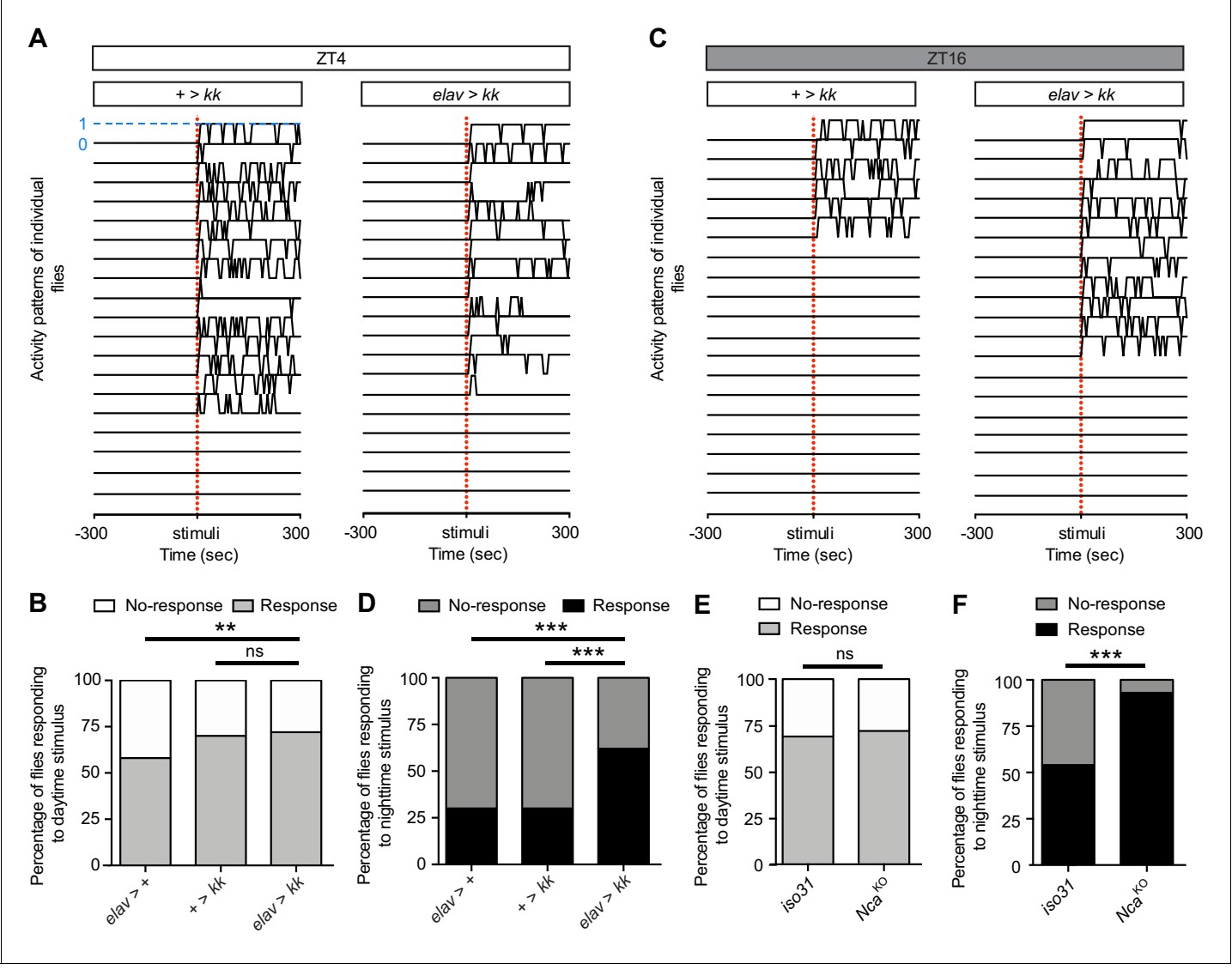

**Figure 2.** NCA reduces responsiveness to stimuli at night under 8L: 16D conditions. (A, C) Locomotor activity in twenty representative control (+ > *kk*) and *Nca*KD (*elav* > *kk*) adult male flies at either ZT4 (A) or ZT16 (C), as measured using the DART system. X-axis denotes 300 s before and after a vibration stimulus (red dotted line). Y-axis represents movement of individual flies in a binary manner (1 = movement, marked by blue dotted line for one fly; 0 = immobility). Only flies that were immobile for five mins preceding the stimulus were selected for analysis. (B, D) Percentage of *Nca*KD and control flies responding or not responding to vibration stimulus at either ZT4 (B) or ZT16 (D). ZT4: *elav* > +: n = 24, + > *kk*: n = 33, *elav* > *kk*: n = 33. ZT16: *elav* > +: n = 23, + > *kk*: n = 30, *elav* > *kk*: n = 29. (E, F) Percentage of *Nca*KO and *iso31* control flies responding or not responding to vibration stimulus at either ZT4 (E) or ZT16 (F). ZT4: *iso31*: n = 48, *Nca*KO: n = 53. ZT16: *iso31*: n = 48, *Nca*KO: n = 44. ns – p>0.05, **p<0.01, ***p<0.001, Binomial test with Bonferonni correction for multiple comparisons.

DOI: https://doi.org/10.7554/eLife.38114.012

The following source data is available for figure 2:

**Source data 1.** Proportion of *Nca* knockdown and knockout flies responding to mechanical stimuli.

DOI: https://doi.org/10.7554/eLife.38114.013

Consistent with this finding, under constant light (LL) conditions, in which the circadian clock becomes arrhythmic due to light-dependent degradation of Timeless (*Hunter-Ensor et al., 1996*; *Koh et al., 2006*; *Peschel et al., 2006*), sleep loss in *Nca*KD flies was completely suppressed (*Figure 3G,H*). From the above data we draw two conclusions. Firstly, that the circadian clock is not required for NCA to regulate sleep per se, but instead defines *when* NCA promotes sleep. Secondly, that light-sensing pathways suppress enhanced wakefulness resulting from reduced NCA expression.

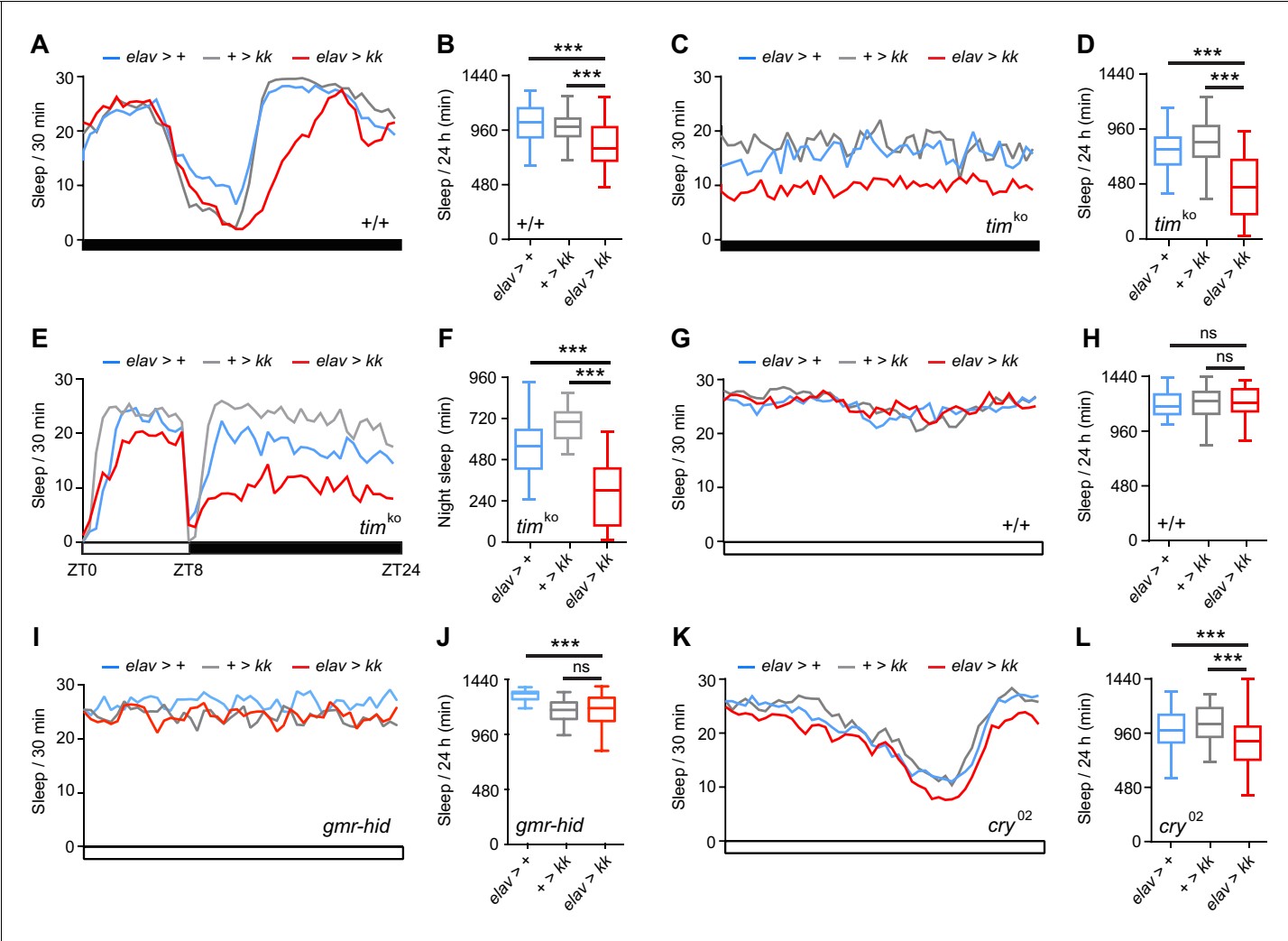

**Figure 3.** Circadian clock and light-sensing pathways define when NCA promotes sleep. (**A–B**) Mean sleep levels in $Nca^{KD}$ and control adult males across 24 hr in constant-dark (DD) conditions (**A**), and total median sleep levels in the above genotypes (**B**). n = 44–47. Note the reduced sleep in the subjective night in $Nca^{KD}$ relative to control adult males, but not the day. (**C–D**) Mean sleep levels in $Nca^{KD}$ and control adult males across 24 hr in DD conditions in a *timeless* knockout ($tim^{KO}$) background (**C**), and total median sleep levels (**D**). n = 32–39. (**E–F**) Mean sleep levels in $Nca^{KD}$ and control adult males across 24 hr in 8L: 16D conditions in a $tim^{KO}$ background (**E**), median night sleep levels (**F**). n = 22–26. (**G–H**) Mean sleep levels in $Nca^{KD}$ and control adult males across 24 hr in constant-light (LL) conditions (**G**), and total median sleep levels (**H**). n = 44–47. (**I–J**) Mean sleep levels in $Nca^{KD}$ and control adult males across 24 hr in LL conditions in a *gmr-hid* background (**I**), and total median sleep levels (**J**). *elav > kk, gmr-hid*/+: n = 51; + > *kk, gmr-hid*/+: n = 48; *elav > +, gmr-hid*/+: n = 24. (**K–L**) Mean sleep levels in $Nca^{KD}$ and control adult males across 24 hr in LL conditions in a *cryptochrome* null ($cry^{02}$) background (**K**), and total median sleep levels in the above genotypes (**L**). n = 61–72. Note the small but consistent reduction in sleep in $Nca^{KD}$, $cry^{02}$ males (**K**), leading to a significant decrease in total median sleep levels relative to controls (**L**). ns - p>0.05, ***p<0.001, as compared to driver and RNAi alone controls via Kruskal-Wallis test with Dunn's post-hoc test.

DOI: https://doi.org/10.7554/eLife.38114.014

The following source data is available for figure 3:

**Source data 1.** Sleep levels in *Nca* knockdown flies under varying environmental and genetic conditions.
DOI: https://doi.org/10.7554/eLife.38114.015

We thus sought to determine which light-sensing pathways restrict sleep loss in $Nca^{KD}$ flies to the night. We reasoned that removing relevant photoreceptive molecules, cells or transduction pathways might restore sleep loss in $Nca^{KD}$ flies during LL. Ablation of photoreceptor cells through expression of the pro-apoptotic gene *hid* (*gmr-hid*) did not alter sleep in $Nca^{KD}$ flies in LL (**Figure 3I,J**). In contrast, using a loss of function allele of *cry* ($cry^{02}$), we found that loss of CRY in LL resulted in a small but significant loss of sleep in $Nca^{KD}$ flies (**Figure 3K,L**). CRY is a blue-light photoreceptor and has

dual roles in synchronization of the circadian clock by light and light-dependent regulation of clock cell excitability (*Fogle et al., 2011*; *Stanewsky et al., 1998*). One or both of these pathways may therefore modulate the timing of sleep loss in $Nca^{KD}$ flies. However, the reduction in sleep in $Nca^{KD}$, $cry^{02}$ flies in LL is lower in magnitude compared to $Nca^{KD}$ flies in DD or 8L: 16D (*Figures 1B* and *3A*), suggesting that additional light-sensing pathways act in concert with CRY to inhibit wakefulness in $Nca^{KD}$ flies in the presence of light. The restoration of clock function in $cry^{02}$ homozygotes in LL may also contribute to sleep loss in $Nca^{KD}$, $cry^{02}$ flies under LL (*Stanewsky et al., 1998*).

## NCA acts in two neuronal subpopulations to promote night sleep

Does NCA act in restricted neuropil regions to promote night sleep? To address this question, we used transgenic RNAi to knock down *Nca* expression in sleep relevant neuronal subpopulations defined by numerous *promoter*-Gal4 driver lines (*Figure 4—figure supplement 1*). These include clock neurons, neurotransmitter- and receptor-specific subtypes, fan-shaped body, mushroom body (MB), mechano-sensory, and visual pathway neurons (*Donlea et al., 2011*; *Guo et al., 2018*; *Jenett et al., 2012*; *Jiang et al., 2016*; *Joiner et al., 2006*; *Lamaze et al., 2018*; *Lamaze et al., 2017*; *Liu et al., 2014*; *Pitman et al., 2006*; *Seidner et al., 2015*; *Sitaraman et al., 2015*). However, in contrast to broadly expressed drivers (*elav*-, *nsyb*- and *inc*-Gal4), *Nca* knockdown in neurotransmitter- or neuropil-specific subsets was insufficient to significantly reduce night sleep (*Figure 4—figure supplement 1A*).

These results suggested that NCA might act in multiple neuropil regions to modulate sleep. Consistent with this hypothesis, we generated a series of driver line combinations and found that *Nca* knockdown using two *enhancer*-Gal4 lines (*R72C01* – an enhancer in the *Dop1R1* locus, and *R14A05* – an enhancer in the *single-minded* locus; we refer to these drivers as *C01* and *A05* respectively) strongly reduced night sleep in 8L: 16D conditions (*Figure 4A,B*) (*Jenett et al., 2012*). *Nca* knockdown using either enhancer alone did not affect night sleep (*Figure 4—figure supplement 2A–D*), nor in combination with dopaminergic, *Dop1R1*-expressing or *cry*-expressing neurons, or components of the anterior visual pathway (*Figure 4—figure supplement 1B*).

The above data indicate that NCA expression in both *C01*- and *A05*-neurons is necessary for normal levels of night sleep. Similarly to pan-neuronal $Nca^{KD}$ flies, knockdown of *Nca* in *C01*- and *A05*-neurons also resulted in sleep loss during the subjective night in DD (*Figure 4C,D*), no sleep loss in LL (*Figure 4E,F*), no alteration in daytime arousal threshold (*Figure 4G*), and a reduced arousal threshold during the night (*Figure 4H*). Thus, we were able to recapitulate the sleep/arousal phenotypes of $Nca^{KD}$ flies by combinatorial *Nca* knockdown in *C01*- and *A05*-neurons.

The *A05* enhancer drives expression in approximately 70 neurons ($70.3 \pm 4.7$, n = 3), as quantified using a fluorescent nuclear marker (*Barolo et al., 2004*). These include a subset of MB Kenyon cells (MB-KCs), a cluster of cell bodies adjacent to the anterior ventrolateral protocerebrum (AVP), and two visual domains: the optic lobe (OL) and anterior optic tubercle (AOTU) (*Figure 5A*). The *C01* enhancer drives expression in approximately 240 neurons ($239.7 \pm 7.8$, n = 3) which encompass MB-KCs as well as neurons projecting to the MB γ-lobes, the antennal mechanosensory and motor center (AMMC), and the superior medial protocerebrum (SMP) (*Figure 5B*). Both drivers label additional cells of unknown identity.

The shared expression of *C01* and *A05* within the MBs raised the possibility that sleep loss in *C01/A05* >*Nca* RNAi flies was due to strong NCA knockdown in neurons labelled by both enhancers. If so, driving *Nca* RNAi with two copies of either *C01* or *A05* should mimic sleep loss in *C01/A05* > *Nca* RNAi flies. However, this was not the case (*Figure 5—figure supplement 1*). Thus, NCA is required in two non-overlapping neuronal populations defined by the *C01* and *A05* enhancers to promote night sleep. Furthermore, since sleep-promoting NCA activity can largely be mapped to approximately 310 neurons but not to wider populations such as cholinergic or GABAergic neurons (*Figure 4—figure supplement 1*), these results argue that sleep loss in *Nca* knockdown and knock-out flies is not simply due to broad neuronal dysfunction.

## NCA functions in the mushroom bodies to regulate sleep and arousal

Detailed examination within MB-KCs using standardized confocal images from the Virtual Fly Brain indicated that the *C01* and *A05* enhancers label non-overlapping regions of the MB, with *C01* expressed in the αβ-KCs, and *A05* expressed in α'β'-KCs (*Figure 5* and *Figure 5—figure*

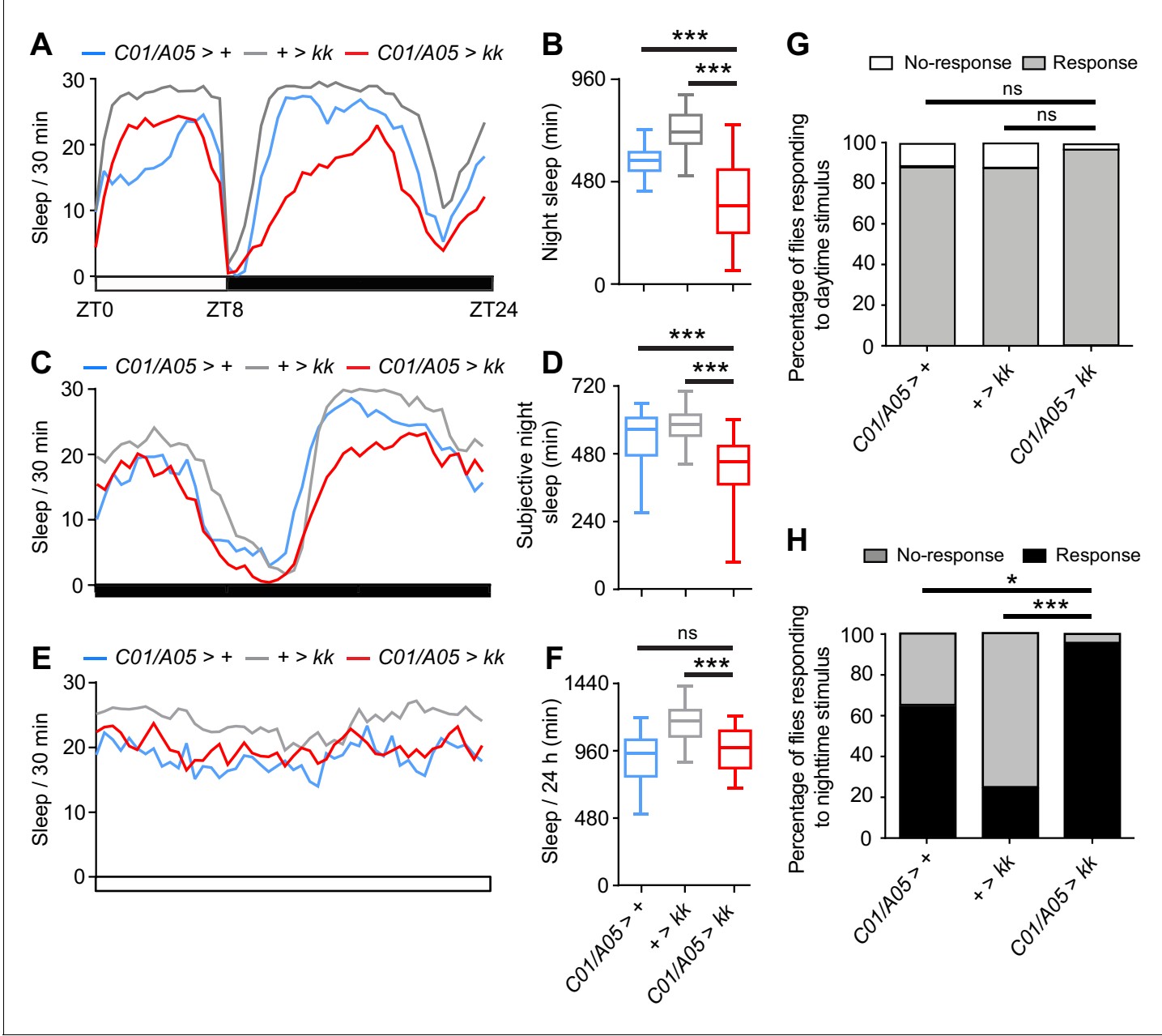

**Figure 4.** NCA acts in a two distinct neural subpopulations to regulate night sleep. (A–F) Sleep patterns in adult male flies with *Nca* knockdown (using the *kk Nca* RNAi) in two neural domains defined by the *A05-* and *C01*-Gal4 drivers in varying light/dark regimes, compared to controls. (A–B) A: mean sleep patterns in 8L: 16D conditions. B: median night sleep in *Nca* knockdown flies compared to heterozygote drivers and transgene alone controls. *+ > kk*: n = 80; *C01/A05 > +*: n = 42; *C01/A05 > kk*: n = 71. (C–D) Mean sleep patterns (C) and median subjective night sleep (D) in constant dark (DD) conditions. *+ > kk*: n = 64; *C01/A05 > +*: n = 47; *C01/A05 > kk*: n = 51. (E–F) Mean sleep patterns (E) and median total sleep (F) in constant light (LL) conditions. *+ > kk*: n = 76; *C01/A05 > +*: n = 26; *C01/A05 > kk*: n = 28. (G–H) Percentage of *C01/A05 > kk* and control flies responding or not responding to vibration stimuli at either ZT4 (G; *C01/A05 > kk*, n = 38, *+ > kk*, n = 61 and *C01/A05 > +*, n = 26) or ZT16 (H; *C01/A05 > kk*, n = 24, *+ > kk*, n = 54 and *C01/A05 > +*, n = 28) under 8L: 16D conditions. ns – p>0.05, *p<0.05, **p<0.01, ***p<0.001, compared to driver and RNAi alone controls, Kruskal-Wallis test with Dunn's post-hoc test (**B, D, F**) or Binomial test with Bonferonni correction for multiple comparisons (**G–H**).
DOI: https://doi.org/10.7554/eLife.38114.016

The following source data and figure supplements are available for figure 4:

**Source data 1.** Sleep levels and proportion of flies responding to mechanical stimuli following *Nca* knockdown in *C01-* and *A05*-neurons or other specific neuronal subtypes, relating to *Figure 4* and associated figure supplements.
DOI: https://doi.org/10.7554/eLife.38114.019

**Figure supplement 1.** Transgenic RNAi-based mini-screen to identify key NCA-expressing neurons.

*Figure 4 continued on next page*

*Figure 4 continued*

DOI: https://doi.org/10.7554/eLife.38114.017
**Figure supplement 2.** *Nca* knockdown in *C01-* or *A05-*neurons alone does not significantly alter sleep.
DOI: https://doi.org/10.7554/eLife.38114.018

*supplement 2A*). Given the known sleep regulatory role of the MB-KCs (*Joiner et al., 2006*; *Pitman et al., 2006*; *Sitaraman et al., 2015*), we examined whether the MB-KCs were an important constituent of either the *C01* and *A05* expression domains.

Similarly to *Nca* knockdown in *C01-* and *A05-*neurons alone (*Figure 4—figure supplement 2A–D*), *Nca* knockdown in the MB-KCs using the *ok107*-Gal4 driver did not alter day or night sleep in 8L: 16D (*Figure 5—figure supplement 2B,C*). However, simultaneous knockdown of *Nca* in *ok107-* and *A05-*neurons significantly reduced night sleep (*Figure 5—figure supplement 2D,E*), whereas *Nca* knockdown in both *ok107-* and *C01-*neurons did not (*Figure 5—figure supplement 2F,G*). Since *Nca* knockdown in *ok107-* and *A05-*neurons partially phenocopies the sleep-inhibiting effect of *Nca* knockdown in *C01-* and *A05-*neurons, these data suggest that the MB-KCs are a relevant component of the *C01* expression domain.

We were also interested to examine whether NCA might act in the MB-KCs to regulate nighttime arousal threshold as well as sleep. Using the DART system, we found that *Nca* knockdown in either *C01-*neurons or in the MB-KCs (using *ok107*-Gal4) significantly increased the number of flies aroused by mechanical stimuli during the night but not the day (*Figure 6A–D*). Since the MB αβ-KCs are labelled by both the *ok107*-Gal4 and *C01*-Gal4 drivers, the above data collectively suggest that NCA acts within the MB αβ-KCs to suppress nocturnal arousal, and that additional circuits within the *A05*-positive domain are required in concert with *C01*-neurons (including MB αβ-KCs) to drive nocturnal hyperactivity when *Nca* expression is reduced.

## NCA inhibits synaptic output in a dark-dependent manner

We next examined whether NCA influences the excitability of *C01-* and *A05-*neurons. To do so, we expressed a genetically encoded fluorescent indicator of neurotransmitter release, UAS-*synapto-pHluorin* (*spH*), in *C01-* and *A05-*neurons with or without *Nca* RNAi. spH localizes to synaptic vesicles and increases in fluorescence in a pH-dependent manner upon vesicle fusion with the presynaptic membrane, providing an optical read-out of neurotransmitter release (*Miesenböck, 2012*). We measured spH fluorescence in four neuropil regions prominently labelled by the *C01-* and *A05*-drivers: the MB αβ-lobes, the antennal mechanosensory and motor center (AMMC), presynaptic innervations of the MB γ-lobes, and the superior medial protocerebrum (SMP). At ZT9-11 in 8L: 16D conditions, *Nca* knockdown in *C01-* and *A05-*neurons resulted in significantly enhanced synaptic release from the MB αβ-lobes and the AMMC (*Figure 7A,B*) but not the MB γ-lobe region or the SMP (*Figure 7C,D*), demonstrating that NCA inhibits synaptic release from a subset of *C01-* and *A05-*neurons and supporting a physiological role for NCA in the MB αβ-lobes.

Since *Nca* knockdown in *C01-* and *A05-*neurons reduces night sleep in 8L: 16D but not in LL conditions (*Figure 4A–B,E–F*), we were interested to test whether the above increases in synaptic release were suppressed in LL. Indeed, at Circadian Time (CT) 9–11 in LL conditions, *Nca* knockdown in *C01-* and *A05-*neurons did not enhance synaptic release from the MB αβ-lobes, the MB γ-lobe region or the SMP, and surprisingly, reduced synaptic release from the AMMC (*Figure 7E–H*). Thus, light-sensing pathways suppress both sleep loss (*Figure 4E,F*) and elevated synaptic release in the MB αβ-lobes and AMMC following *Nca* knockdown in *C01-* and *A05-*neurons.

## NCA acts in wake-promoting neurons

Our results suggested a model in which loss of NCA causes aberrant excitation of a neural network that promotes wakefulness in the absence of light. This model yields two predictions. Firstly, that artificial activation of *C01-* and *A05-*neurons should promote wakefulness. Secondly, that reducing excitability of *C01-* and *A05-*neurons should suppress sleep loss in *Nca* knockdown flies.

To test our first prediction, we stimulated *C01-* and *A05-*neurons by expressing the temperature-sensitive cation channel TrpA1 in either neuronal subset or both and shifting flies from a non-activating temperature (22°C) to an activating temperature (27°C) (*Hamada et al., 2008*) (*Figure 8A*). At

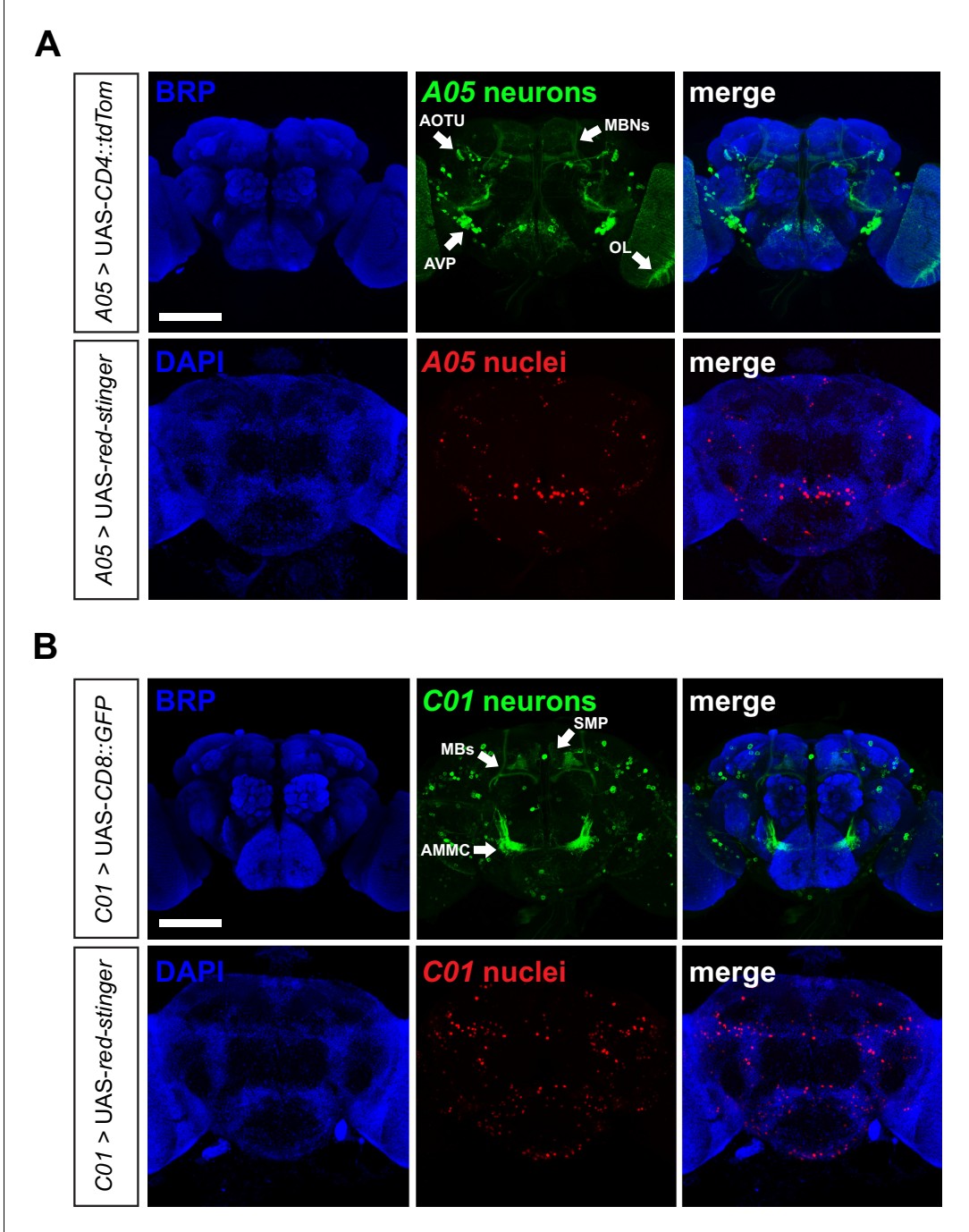

**Figure 5.** Distribution of *A05*- and *C01*-neurons in the adult *Drosophila* brain. (**A–B**) Confocal z-stacks of adult male brains expressing genetically-encoded fluorophores labelling either neuronal processes (CD4::TdTom or CD8::GFP) or nuclei (Red-stinger) under the *A05*- (**A**) or *C01*-Gal4 (**B**) drivers. Neuropil regions are labelled with anti-Bruchpilot (BRP). Nuclei are co-labelled with DAPI. Scale bars, 100 µm. Arrows point to neuropil centers. AOTU: anterior optic tubercle. MBNs: mushroom body neurons. OL: optic lobe. AMMC: antennal mechanosensory and motor center. AVP: anterior ventrolateral protocerebrum. SMP: superior medial protocerebrum.

DOI: https://doi.org/10.7554/eLife.38114.020

The following source data and figure supplements are available for figure 5:

**Source data 1.** Sleep levels following *Nca* knockdown in *C01*-, *A05*- or *ok107*-neurons (or combinations of), relating to *Figure 5—figure supplements 1* and *2*.

DOI: https://doi.org/10.7554/eLife.38114.023

**Figure supplement 1.** *Nca* knockdown using homozygous *C01*- or *A05*-Gal4 drivers does not affect night sleep.

*Figure 5 continued on next page*

*Figure 5 continued*

DOI: https://doi.org/10.7554/eLife.38114.021

**Figure supplement 2.** The mushroom bodies are a sleep-relevant subdomain within *C01*-neurons.

DOI: https://doi.org/10.7554/eLife.38114.022

the non-activating temperature, over-expression of TrpA1 in either neuronal population or both did not affect sleep (*Figure 8B*). At the activating temperature, excitation of *A05*-neurons did not alter night sleep (*Figure 8C,D*). In contrast, excitation of *C01*-neurons profoundly reduced night sleep (*Figure 8C,D*) as well as day sleep (*Figure 8C*). Interestingly, simultaneous activation of *C01*- and *A05*-neurons further reduced night but not day sleep relative to activation of *C01*-neurons alone, despite activation of *A05*-neurons alone having no impact on sleep in 8L: 16D (*Figure 8C,D*). *C01*- and *A05*-neurons thus synergistically interact to modulate night sleep.

To test our second prediction, we over-expressed a non-inactivating outward rectifying potassium channel (dORKΔC2) in *C01*- and *A05*-neurons with and without *Nca* knockdown via RNAi. Here, expression of dORKΔC2 is predicted to suppress neuronal firing by hyperpolarizing the resting membrane potential (*Nitabach et al., 2002*; *Park and Griffith, 2006*). Silencing *C01*- and *A05*-neurons with dORKΔC2 in an otherwise wild type background did not alter day or night sleep levels (*Figure 8E,F*; p>0.99 compared to *dORKΔC2/+*controls, Kruskal-Wallis test with Dunn's post-hoc test). However, consistent with the above prediction, expression of dORKΔC2 in concert with *Nca* RNAi significantly suppressed night sleep loss relative to male flies expressing *Nca* RNAi alone or alongside an innocuous transgene (UAS-*FRT-stop-FRT-GFP*) (p<0.0005). Thus, NCA promotes night sleep by limiting synaptic output from arousal- and wake-promoting neurons within the *C01*- and *A05*-Gal4 domains that include the MB αβ-KCs.

## Discussion

Human sleep can be partitioned into stages characterized by unique electroencephalographic signatures and differing arousal thresholds (*Rechtschaffen et al., 1966*; *Rechtschaffen and Kales, 1968*). Across the day/night cycle, *Drosophila* sleep is similarly characterized by dynamic alterations in arousal threshold, with day sleep associated with lower arousal thresholds relative to night sleep (*Faville et al., 2015*; *van Alphen et al., 2013*). However, molecular pathways underlying distinct sleep stages are poorly defined. Here we demonstrate a role for the neuronal calcium sensor NCA as a regulator of nocturnal sleep and arousal, thus providing a novel entry point to address this issue.

Previous genetic screens have identified an array of sleep-promoting factors in *Drosophila* (*Tomita et al., 2017*). However, despite extensive circuit analyses, the complete neural substrates in which these factors function have yet to be determined (*Afonso et al., 2015*; *Rogulja and Young, 2012*; *Shi et al., 2014*; *Stavropoulos and Young, 2011*; *Tomita et al., 2015*; *Wu et al., 2014*). Our results are consistent with these findings and offer a tentative explanation for the difficulties in defining circuit requirements for sleep-relevant proteins in *Drosophila*. We show that NCA is not required within a single cell-type or neuropil region to inhibit nighttime arousal and wakefulness. Instead, sleep-relevant NCA activity is necessary within two distinct domains of the *Drosophila* nervous system defined by the *A05*- and *C01*-Gal4 drivers (*Jenett et al., 2012*).

Ex vivo imaging demonstrates that *Nca* knockdown enhances synaptic output from subsets of *C01*- and *A05*-neurons innervating the MB αβ-lobes and the AMMC. Reversing this effect via dORKΔC2-mediated electrical silencing suppresses sleep loss in *Nca* knockdown flies, suggesting that enhanced synaptic output from *C01*- and *A05*-neurons via drives nighttime wakefulness. We note that while dORKΔC2 expression does not grossly effect the development or axonal guidance of particular clock neurons in *Drosophila* (*Nitabach et al., 2002*), prior work has shown that potassium channel overexpression can reduce the viability of mammalian hippocampal neurons (*Nadeau et al., 2000*). Thus, we cannot entirely rule out an effect of dORKΔC2 expression that is secondary to electrical silencing. However, adult-stage excitation via heat-activated TrpA1 channels reveals a clear capacity of *C01*-neurons to promote wake during both day and night, whereas *A05*-neurons promote nighttime wakefulness only when *C01*-neurons are concurrently activated. Since this thermogenetic approach avoids unforeseen effects of chronic alterations in excitability on cellular processes

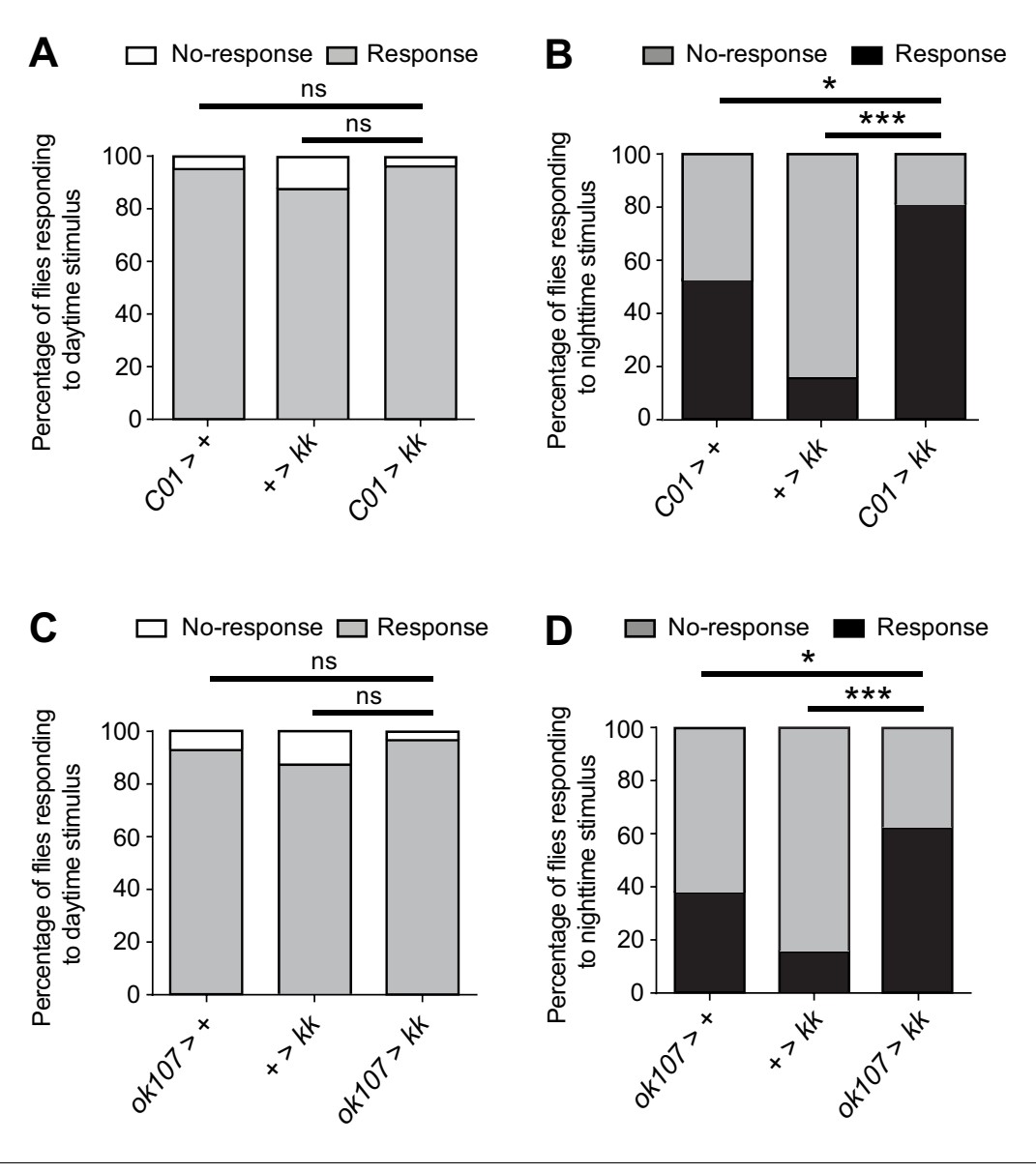

**Figure 6.** NCA acts in the mushroom bodies to regulate nocturnal arousal. (A–B) Percentage of adult male flies expressing *Nca* RNAi (*kk*) in *C01*-neurons (*C01 > kk*) and control flies responding or not responding to vibration stimulus at either ZT4 (day; A) or ZT16 (night; B). ZT4: *C01 > +*, n = 22, *+ > kk*, n = 61, *C01 > kk*, n = 27. ZT16: *C01 > +*, n = 19, *+ > kk*, n = 54, *C01 > kk*, n = 21. (C–D) Percentage of adult male flies expressing *Nca* RNAi (*kk*) in MB-KCs (*ok107 > kk*) and control flies responding or not responding to vibration stimulus at either ZT4 (day; C) or ZT16 (night; D). ZT4: *ok107 > +*, n = 26, *+ > kk*, n = 47, *ok107 > kk*, n = 28. ZT16: *ok107 > +*, n = 26, *+ > kk*, n = 44, *ok107 > kk*, n = 27. ns – p>0.05, *p<0.05, ***p<0.001, Binomial test with Bonferonni correction for multiple comparisons.

DOI: https://doi.org/10.7554/eLife.38114.024

The following source data is available for figure 6:

**Source data 1.** Proportion of flies responding to mechanical stimuli following *Nca* knockdown in *C01*- or *ok107*-neurons.

DOI: https://doi.org/10.7554/eLife.38114.025

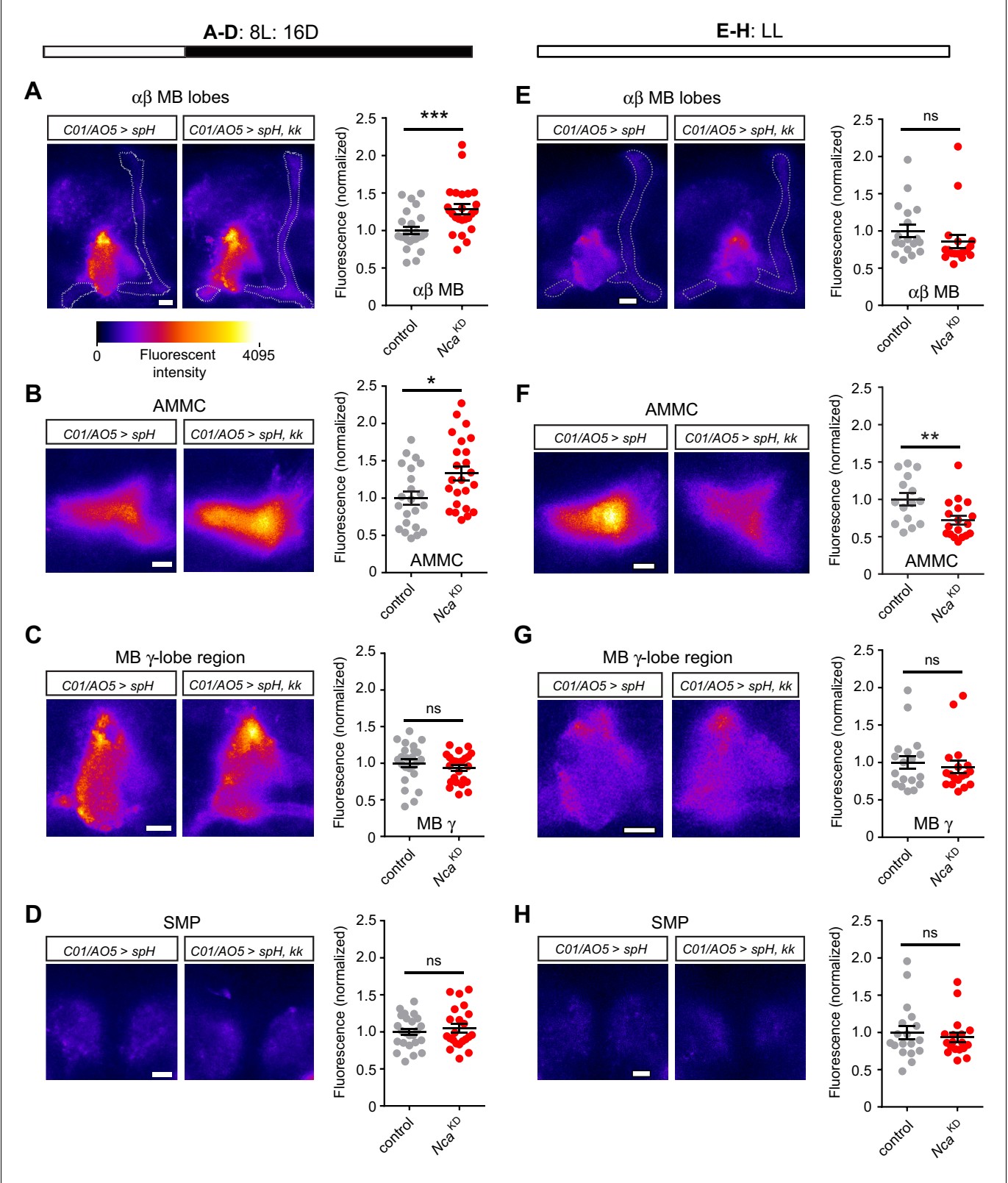

**Figure 7.** NCA suppresses synaptic release in subsets of *C01/A05*-neurons during darkness. (**A–D**) Fluorescence of an optical reporter of synaptic release (synapto-pHluorin, spH) in neuropil regions labelled by the *C01-* and *A05*-drivers, in control adult males (*C01/A05 > spH*) or following *Nca* knockdown in *C01-* and *A05*-neurons (*C01/A05 > spH, kk*). Flies were housed under 8L: 16D conditions, in which *Nca* knockdown in *C01-* and *A05*-neurons causes robust nighttime sleep loss. (**E–G**) spH fluorescence in control adult males or following *Nca* knockdown in *C01-* and *A05*-neurons (*C01/*

*Figure 7 continued*

*A05 > spH, kk*). Flies were housed in LL conditions, in which *Nca* knockdown in *C01-* and *A05*-neurons has no effect on sleep levels. In each panel, representative confocal images of spH fluorescence (left) and mean fluorescent intensity (right, normalized to the mean of *C01/A05 > spH* controls) are shown. Dots within dot plots represent individual brain hemisphere measurements. A-D: n = 22–24. E-H: n = 15–18. Neuropil regions are noted. MB: mushroom body. AMMC: antennal mechanosensory motor center. SMP: superior medial protocerebrum. ns – p>0.05, *p<0.05, **p<0.01, ***p<0.001, Mann-Whitney U-test.

DOI: https://doi.org/10.7554/eLife.38114.026

The following source data is available for figure 7:

**Source data 1.** Normalized synaptopHluorin fluorescence in specified neuropil regions (see *Figure 7*) in a wild-type background or following *Nca* knockdown in *C01-* and *A05*-neurons, in either 8L: 16D or in constant light (LL).
DOI: https://doi.org/10.7554/eLife.38114.027

(*Depetris-Chauvin et al., 2011*), the above data collectively support a model in which reduced NCA activity in *C01-* and *A05*-neurons causes a mild elevation in neurotransmitter release from neuronal subsets within the *C01-* and *A05*-domains. Reduced NCA activity in *C01-* or *A05*-neurons alone is insufficient to promote wakefulness. Yet when NCA expression is inhibited in *C01-* and *A05*-neurons simultaneously, the resulting enhancement of synaptic output within this wider network is sufficient to reduce night sleep.

While the precise identities of the wake-promoting circuits within the *C01-* and *A05*-domains remain enigmatic, our data suggests a role for NCA in the MB αβ-lobes in suppressing arousal during the night. The MB-KCs have been shown to exert a multifaceted influence on *Drosophila* sleep (*Joiner et al., 2006*; *Pitman et al., 2006*; *Sitaraman et al., 2015*). Recent data has shown that thermo-genetic activation of MB αβ-lobes does not affect sleep levels (*Sitaraman et al., 2015*). Similarly, we find that *Nca* knockdown in MB-KCs or in *C01*-neurons (which overlap in the MB αβ-lobes) does not impact sleep in 8L: 16D. Nonetheless, either manipulation is sufficient to reduce the arousal threshold in the context of a mechanical stimulus. Thus, NCA plays dual functions in modulating arousal and wakefulness, likely by acting in distinct circuits within the fly brain.

Two questions arise from these results. Firstly, how might NCA inhibit synaptic output? The mammalian NCA homolog Hippocalcin modulates neuronal excitability and plasticity through multiple pathways. Hippocalcin facilitates NMDA receptor endocytosis during LTD and gates the slow afterhyperpolarisation, a calcium-activated potassium current controlling spike frequency adaptation (*Andrade et al., 2012*; *Jo et al., 2010*; *Tzingounis et al., 2007*). Recent data suggest that Hippocalcin also inhibits calcium influx through N- and P/Q-type voltage-gated calcium channels (*Helassa et al., 2017*). Given the strong homology between Hippocalcin and NCA, it will be intriguing to test whether NCA limits excitatory synaptic input and reduces spike frequency and/or neurotransmitter release through similar pathways in *Drosophila*. Indeed, cell-type-specific expression of homologous NCA-binding proteins may explain why synaptic output is enhanced in only a subset of *C01-* and *A05*-neurons following *Nca* knockdown, despite previous results showing that NCA is broadly expressed in the *Drosophila* brain (*Teng et al., 1994*).

Secondly, how is the sleep-promoting role of NCA limited to the night? Our results show that both internal and external cues regulate when NCA impacts sleep. *Nca* knockdown reduces sleep solely during the subjective night in DD, but throughout 24 hr in DD when the circadian clock is disrupted. Thus, our data demonstrate a role for the clock in timing when NCA promotes sleep. However, light also acts in parallel as an environmental signal capable of suppressing enhanced wakefulness when NCA activity is reduced, in part through the CRY photoreceptor. At the circuit-level, our results suggest that constant light suppresses increased neurotransmitter release from neurons in the MB αβ-lobes and AMMC following *Nca* knockdown, further supporting a role for the MB αβ-KCs as a component of the neural network through which NCA influences sleep and suggesting a potential contribution from neurons innervating the AMMC. Elucidating the identity of clock- and light-regulated circuits (including CRY-expressing neurons) that gate when and whether NCA promotes sleep will prove a fruitful avenue of future research. More broadly, our work provides a framework to study how complex interactions between genes, neural circuits and the environment influence a critical behavior such as sleep.

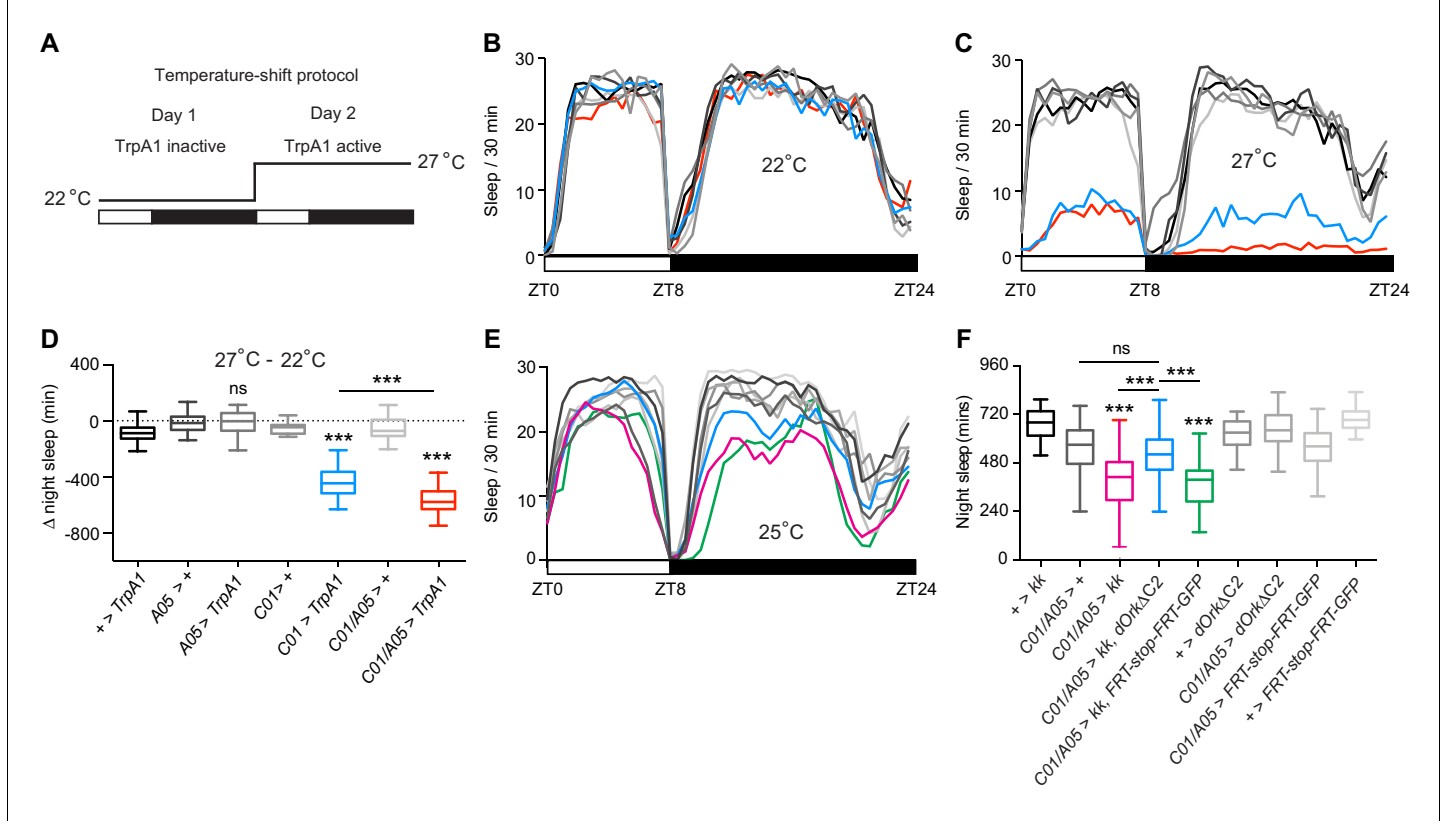

**Figure 8.** Sleep loss in *Nca* knockdown flies is caused by enhanced excitability of *C01/A05*-neurons. (**A**) Experimental paradigm for acute activation of *A05* or *C01*-neurons. 22°C: non-activating temperature for TrpA1. 27°C: activating temperature. Sleep levels were recorded over two days in 8L: 16D conditions. (**B–C**) Mean sleep levels across 8L: 16D following expression of TrpA1 in *A05*-, *C01*- or *A05*- and *C01*-neurons (and associated controls) at 22°C (**B**) or 27°C (**C**). (**D**) Median change in night sleep levels (Δ night sleep) following the shift from 22°C on day 1°C to 27°C on day 2. + > *TrpA1*: n = 53, *A05* > +: n = 23, *A05* > *TrpA1*: n = 68, *C01* > +: n = 24, *C01* > *TrpA1*: n = 40, *C01/A05* > +: n = 33, *C01/A05* > *TrpA1*: n = 40. ns – p>0.05, ***p<0.001, as compared to *TrpA1* or driver alone controls by Kruskal-Wallis test with Dunn's post-hoc test (for *C01* > *TrpA1*, *A05* > *TrpA1*, or *C01/A05* > *TrpA1* compared to controls) or Mann-Whitney U-test (for *C01/A05* > *TrpA1* compared to *C01* > *TrpA1*). (**E–F**) Inhibition of *C01/A05*-neurons by expressing dORKΔC2 rescues sleep loss due to *Nca* knockdown, while expression of dORKΔC2 does not change baseline sleep. Mean sleep patterns in 8L: 16D conditions are shown in (**E**). Median night sleep levels are shown in (**F**).+ > *kk*: n = 72, *C01/A05* > +: n = 85, *C01/A05* > *kk*: n = 95, *C01/A05* > *dORKΔC2, kk*: n = 77, *C01/A05* > *kk, FRT-stop-FRT-GFP*: n = 39, + > *dORKΔC2*: n = 57, *C01/A05* > *dORKΔC2*: n = 73, *C01/A05* > *FRT-stop-FRT-GFP*: n = 49, + > *FRT-stop-FRT-GFP*: n = 36. ns – p>0.05, ***p<0.001, Kruskal-Wallis test with Dunn's post-hoc test.
DOI: https://doi.org/10.7554/eLife.38114.028

The following source data is available for figure 8:

**Source data 1.** Sleep levels following excitation or inhibition of *C01*- and *A05*-neurons (simultaneously or in isolation), either in a wild type background or in parallel to *Nca* knockdown.
DOI: https://doi.org/10.7554/eLife.38114.029

# Materials and methods

## Key resources table

| Reagent type (species) or resource | Designation | Source or reference | Identifiers | Additional information |
|---|---|---|---|---|
| Genetic reagent (*Drosophila melanogaster*) | kk108825 | Vienna *Drosophila* Resource Center | RRID:FlyBase_FBst0481000 | |
| Genetic reagent (*Drosophila melanogaster*) | y[1]v[1]; P{y[+t7.7] v[+t1.8]=TRiP.HMJ21533}attP40 | Bloomington Stock Center | RRID:BDSC_54814 | |

*Continued on next page*

Continued

| Reagent type (species) or resource | Designation | Source or reference | Identifiers | Additional information |
|---|---|---|---|---|
| Genetic reagent (Drosophila melanogaster) | y[1] v[1]; P{y[+t7.7] v[+t1.8]=TRiP.JF03398}attP2 | Bloomington Stock Center | RRID:BDSC_29461 | |
| Genetic reagent (Drosophila melanogaster) | w[*]; P{w[+mC]=ple-GAL4.F}3 | Bloomington Stock Center | RRID:BDSC_8848 | |
| Genetic reagent (Drosophila melanogaster) | w[1118];P{w[+mC]=ChAT-GAL4.7.4}19B/CyO, P{ry[+t7.2]=sevRas1 .V12}FK1 | Bloomington Stock Center | RRID:BDSC_6798 | |
| Genetic reagent (Drosophila melanogaster) | w[1118]; P{w[+mW.hs]=GawB}VGlut[OK371] | Bloomington Stock Center | RRID:BDSC_26160 | |
| Genetic reagent (Drosophila melanogaster) | P{w[+mC]=Gad1 GAL4.3.098}2/CyO | Bloomington Stock Center | RRID:BDSC_51630 | |
| Genetic reagent (Drosophila melanogaster) | w[1118]; P{w[+mC]=Ddc-GAL4.L}4.3D | Bloomington Stock Center | RRID:BDSC_7010 | |
| Genetic reagent (Drosophila melanogaster) | w[*]; P{w[+mC]=GAL4 ninaE.GMR}12 | Bloomington Stock Center | RRID:BDSC_1104 | |
| Genetic reagent (Drosophila melanogaster) | w[1118]; P{w[+mC]=Trh-GAL4.long}2 | Bloomington Stock Center | RRID:BDSC_38388 | |
| Genetic reagent (Drosophila melanogaster) | w[*]; P{w[+mC]=Tdc2 GAL4.C}2 | Bloomington Stock Center | RRID:BDSC_9313 | |
| Genetic reagent (Drosophila melanogaster) | w[*]; P{w[+mW.hs]=GawB}cv-c[C5] | Bloomington Stock Center | RRID:BDSC_30839 | |
| Genetic reagent (Drosophila melanogaster) | w[*]; P{w[+mW.hs]=GawB}OK107 ey[OK107]/In(4)ci[D], ci[D] pan[ciD] sv[spa-pol] | Bloomington Stock Center | RRID:BDSC_854 | |
| Genetic reagent (Drosophila melanogaster) | y[1] w[1118]; PBac{w[+mC]=5HPw[+]}Nca[A502] | Bloomington Stock Center | RRID:BDSC_16130 | |
| Genetic reagent (Drosophila melanogaster) | w[1118]; P{y[+t7.7] w[+mC]=GMR23E10-GAL4}attP2 | Bloomington Stock Center | RRID:BDSC_49032 | |
| Genetic reagent (Drosophila melanogaster) | w[1118]; P{y[+t7.7] w[+mC]=GMR55B01-GAL4}attP2 | Bloomington Stock Center | RRID:BDSC_39100 | |
| Genetic reagent (Drosophila melanogaster) | w[1118]; P{y[+t7.7] w[+mC]=GMR52 H12-GAL4}attP2 | Bloomington Stock Center | RRID:BDSC_38856 | |
| Genetic reagent (Drosophila melanogaster) | w[1118]; P{y[+t7.7] w[+mC]=GMR17 F12-GAL4}attP2 | Bloomington Stock Center | RRID:BDSC_48779 | |
| Genetic reagent (Drosophila melanogaster) | w[1118]; P{y[+t7.7] w[+mC]=GMR72B05-GAL4}attP2 | Bloomington Stock Center | RRID:BDSC_39611 | |
| Genetic reagent (Drosophila melanogaster) | w[1118]; P{y[+t7.7] w[+mC]=GMR72B07-GAL4}attP2 | Bloomington Stock Center | RRID:BDSC_39764 | |

*Continued*

| Reagent type (species) or resource | Designation | Source or reference | Identifiers | Additional information |
|---|---|---|---|---|
| Genetic reagent (*Drosophila melanogaster*) | w[1118]; P{y[+t7.7] w[+mC]=GMR72B08-GAL4}attP2 | Bloomington Stock Center | RRID:BDSC_46669 | |
| Genetic reagent (*Drosophila melanogaster*) | w[1118]; P{y[+t7.7] w[+mC]=GMR72 C01-GAL4}attP2 | Bloomington Stock Center | RRID:BDSC_41358 | |
| Genetic reagent (*Drosophila melanogaster*) | w[1118]; P{y[+t7.7] w[+mC]=GMR72 C01-GAL4}attP2 | Bloomington Stock Center | RRID:BDSC_47729 | |
| Genetic reagent (*Drosophila melanogaster*) | w[1118]; P{y[+t7.7] w[+mC]=GMR72 C02-GAL4}attP2/TM3, Sb[1] | Bloomington Stock Center | RRID:BDSC_46672 | |
| Genetic reagent (*Drosophila melanogaster*) | w[1118]; P{y[+t7.7] w[+mC]=GMR78B07-GAL4}attP2 | Bloomington Stock Center | RRID:BDSC_39989 | |
| Genetic reagent (*Drosophila melanogaster*) | w[1118]; P{y[+t7.7] w[+mC]=GMR88A06-GAL4}attP2 | Bloomington Stock Center | RRID:BDSC_46847 | |
| Genetic reagent (*Drosophila melanogaster*) | w[1118]; P{y[+t7.7] w[+mC]=GMR91A07 -GAL4}attP2/TM3, Sb[1] | Bloomington Stock Center | RRID:BDSC_47147 | |
| Genetic reagent (*Drosophila melanogaster*) | *cg7674* RNAi 1 (chromosome III) | NIG-FLY stock center | Accession number: NM_140910.2 | |
| Genetic reagent (*Drosophila melanogaster*) | *cg7674* RNAi 2 (chromosome II) | NIG-FLY stock center | Accession number: NM_140910.2 | |
| Genetic reagent (*Drosophila melanogaster*) | nompC-Gal4 | *Kamikouchi et al., 2009* | | |
| Genetic reagent (*Drosophila melanogaster*) | inc-Gal4:2 | *Stavropoulos and Young, 2011* | | |
| Genetic reagent (*Drosophila melanogaster*) | ppk-Gal4 | *Zhong et al., 2012* | | |
| Genetic reagent (*Drosophila melanogaster*) | TrpA1-CD-Gal4 | *Zhong et al., 2012* | | |
| Genetic reagent (*Drosophila melanogaster*) | tim^KO | *Lamaze et al., 2018* | | |
| Genetic reagent (*Drosophila melanogaster*) | GMR14A05-Gal4 | Janelia Research Campus FlyLight Project | 26432 | |
| Genetic reagent (*Drosophila melanogaster*) | w[1118];+; *Nca*[ko1]/TM2 | This paper | | Null allele of *Nca* |
| Genetic reagent (*Drosophila melanogaster*) | w[1118];+; *Nca*[ko2]/TM2 | This paper | | *Nca* null allele (second allele) |
| Genetic reagent (*Drosophila melanogaster*) | w[1118];+; *Nca*[ko3]/TM2 | This paper | | *Nca* null allele (third allele) |

*Continued*

| Reagent type (species) or resource | Designation | Source or reference | Identifiers | Additional information |
|---|---|---|---|---|
| Strain, strain background (*Drosophila melanogaster*) | Canton-S | Bloomington Stock Center | RRID:BDSC_64349 | |
| Antibody | Rabbit anti-DsRed | Clontech | RRID:AB_10013483 | (1:2000) |
| Antibody | Mouse anti-Bruchpilot | Developmental Studies Hybridoma Bank | RRID:AB_2314866 | (1:200) |
| Antibody | Rabbit anti-GFP | Invitrogen | RRID:AB_221569 | (1:1000) |
| Antibody | Goat anti-Mouse Alexa Fluor-647 | ThermoFisher | RRID:AB_141725 | (1:500) |
| Antibody | Alexa Fluor 488 goat anti-rabbit IgG | ThermoFisher | RRID:AB_2576217 | (1:2000) |
| Antibody | Alexa Fluor 555 goat anti-rabbit IgG | ThermoFisher | RRID:AB_2633281 | (1:2000) |
| Antibody | DAPI | Sigma-Aldrich | D9542-10MG | |
| Commercial assay or kit | Wizard SV Gel and PCR Clean-Up System | Promega | Cat. #: A9281 | |
| Commercial assay or kit | Zero Blunt T OPO PCR Cloning Kit | ThermoFisher Scientific | Cat. #: 450245 | |
| Commercial assay or kit | TRIzol | ThermoFisher Scientific | Cat. #: 15596026 | |
| Commercial assay or kit | MMLV RT | Promega | Cat. #: M170A | |
| Commercial assay or kit | Power SYBR Green Master Mix | ThermoFisher Scientific | Cat. #: 4367659 | |

## Fly husbandry

Flies were maintained on standard fly food at constant temperature 25°C under 12 hr: 12 hr light-dark cycles (12L: 12D). The following strains were obtained from the Bloomington, VDRC and NIG-FLY stock centers: kk108825 (100625), hmj21533 (54814), jf03398 (29461), *ple*-Gal4 (8848), *Chat*-Gal4 (6798), *vGlut*-Gal4 (26160), *GAD*-Gal4 (51630), *Ddc*-Gal4 (7010), *GMR*-Gal4 (1104), *Trh.1*-Gal4 (38388), *Tdc2*-Gal4 (9313), *C5*-Gal4 (30839), *ok107*-Gal4 (854), $Nca^{A502}$ (16130), *cg7646* RNAi 1 (7646R-1) and *cg7646* RNAi 2 (7646R-2). The remaining lines obtained from the Bloomington stock center are part of the Janelia Flylight collection with identifiable prefixes: R23E10-Gal4, R55B01-Gal4, R52H12-Gal4, Hdc-Gal4 (R17F12-Gal4), R14A05-Gal4, R72B05-Gal4, R72B07-Gal4, R72B08-Gal4, R72B11-Gal4, R72C01-Gal4, R72C02-Gal4, R78B07-Gal4, R91A07-Gal4, and R88A06-Gal4. The following lines were generous gifts from Kyunghee Koh: *elav*-Gal4, *nsyb*-Gal4, *tim*-Gal4, *TUG*-Gal4 and *cry*-Gal4:16; Joerg Albert: *nompC*-Gal4 (*Kamikouchi et al., 2009*) and Nicolas Stavropouplos: *inc*-Gal4:2 (*Stavropoulos and Young, 2011*). *ppk*-Gal4 and *TrpA1*-CD-Gal4 were described previously (*Zhong et al., 2012*). GMR-hid, $tim^{KO}$ and $cry^{02}$ were previously described in *Lamaze et al. (2017)*. Except for *Ddc*-Gal4, *Trh.1*-Gal4, *Tdc2*-Gal4, *nompC*-Gal4 and *Hdc*-Gal4, all *Drosophila* strains above were either outcrossed five times into an isogenic control background (*iso31*) or insertion-free chromosomes were exchanged with the *iso31* line (*hmj21533 and jf03398*) before testing for sleep-wake activity behavior. Note: R14A05-Gal4 was initially mislabelled as R21G01-Gal4 in the Bloomington shipment. The mismatch between the image of R21G01 > GFP in the FlyLight database and our immuno-staining data (*A05*, *Figure 5A*) led us to clarify the actual identity of the line as R14A05-Gal4 by sequencing genomic PCR product using the following primers pair: pBPGw_ampF: agggttattgtctcatgagcgg and pBPGw_Gal4R: ggcgcacttcggtttttctt.

## Generation of *Neurocalcin* knockout alleles

Null alleles of *Nca* were generated using homologous recombination as described previously (*Baena-Lopez et al., 2013*). Briefly, genomic DNA was extracted from 20 wild type flies (Canton S)

using the BDGP buffer A-LiCl/KAc precipitation protocol (http://www.fruitfly.org/about/methods/inverse.pcr.html). The 5' (Arm 1) and 3' (Arm 2) genomic regions flanking the *Nca* coding sequence were PCR amplified via high fidelity DNA polymerase (Q5 high-fidelity 2X master mix, M0492S, NEB) with the following primers: NotI_Arm1F1: gcggccgctaatttgcagctctgcatcg, NotI_Arm1R1: gcggccgcatggtaagaagcacgcaacc, AscI_Arm2F1: ggcgcgccttatgaccgttccaaaacacc, AvrII_Arm2R1: cctaggggctaaatacgttgaccaagc. The corresponding Arm1 and Arm2 fragments (~2.5 kb) were gel purified (Wizard SV Gel and PCR Clean-Up System, A9281, Promega) and cloned into pCR-Blunt II-TOPO vector (Zero Blunt TOPO PCR Cloning Kit, 450245, ThermoFisher Scientific), and subsequently sub-cloned via NotI (R3189S, NEB) and AscI/AvrII digestion (R0558S and R0174S, NEB) and T4 ligation (M0202S, NEB) into the pTV$^{cherry}$ vector, a P-element construct containing the mini-*white*$^+$ marker and UAS-*reaper* flanked by FRT and I-SceI sites (*Baena-Lopez et al., 2013*). The sequence identifies of Arm one and Arm two fragments within the pTV$^{cherry}$ vector were verified via Sanger sequencing using the following primers: nca1_f: cagctctgcatcgctttttgt, nca1_3_f: ccctcgcgcatggtacttta, nca1_r: agcgtcacataagttctccca, nca1_4_f: tggacgaaaataacgatggtca, nca1_5_f: agactacttagccatgttttcatact, nca1_2_f: tgacgaagccacaattaaagagtg, nca1_1_f: gcaaccctgttcccctttca, nca2_f: gaccgttccaaaacaccca, nca2_3_f: ttgttgtgcgccacgtttc, nca2_r: acgtatgctccatgattcctct nca2_4_f: tgcaggtcggttaatcaatgc, nca2_5_f: tcaatcgatttggggccagg, nca2_2_f: ccttctccaggctcagcaaa, nca2_1_f: actctgcatttcgataagattagcc. Donor lines containing the pTV vector with Arm1 and Arm2 homologous fragments (pTV_nca1 + 2) were then generated via embryonic injection and random P-element mediated genomic insertions (Bestgene inc CA, USA). To initiate homologous recombination between pTV_nca1 + 2 and the endogenous *Nca* locus, donor lines were crossed to *yw*; hs-*flp*, hs-I-*SceI*/CyO and the resulting larvae were heat shocked at 48 hr and 72 hr after egg laying for 1 hr at 37°C. Around 200 female offspring with mottled/mosaic red eyes were crossed in pools of three to *ubiquitin*-Gal4[3xP3-GFP] males to remove nonspecific recombination events (via UAS-*reaper*-mediated apoptotic activity). The crossings were flipped once over and the progeny (~12000 adults) was screened for the presence of red-eyed and GFP-positive flies. Three independent GFP$^+$ red-eyed lines (*ko1*, *ko2*, and *ko3*) were identified. The exchange of endogenous *Nca* locus with pTV_nca1 + 2 fragments was confirmed by detecting a 2.6 kb PCR product (*Figure 1—figure supplement 1C*) in the genomic DNA samples of the above three lines (pre-digested by EcoRI/NotI) using the following primer pairs: ncaKO-F2: tgggaattgactgatacagcct; ncaKO-R2: ggcactacggtacctg-cat. ncaKO-F2 matches to the region between 24 bp and 2 bp upstream of Arm1 and ncaKO-R2 overlaps with attP site (*Figure 1A*). The absence of endogenous *Nca* mRNA in *ko1* flies was confirmed by standard and quantitative RT-PCR (*Figure 1—figure supplement 5D,E*; also see below). The min-*white*$^+$ cassette and majority of pTV vector sequences were further removed from the *ko1* genome via Cre-loxP recombination (*Figure 1A*). This 'Cre-out' strain was then backcrossed five times to a *Nca*$^{A502}$ line (where *A502* is a P-element insertion two kbp upstream of the *Nca* CDS) that was outcrossed previously into the *iso31* background (see Fly husbandry section). Before testing for changes in sleep/wake behaviour, the resulting line, termed *Nca* knockout (*Nca*$^{KO1}$), was lastly verified by sequencing a 576 bp genomic PCR product (using primer pair: nca1_5_f and nca2_r), confirming the absence of *Nca* CDS sequence and the insertion of an attP site in the *Nca* locus. Two independent 'Cre-out' lines derived from the *ko2* and *ko3* alleles were also outcrossed to *Nca*$^{A502}$ for two generations (*Nca*$^{KO2}$ and *Nca*$^{KO3}$) and tested for sleep-wake behaviour.

## RNA extraction and quantitative PCR

For RNA extractions, 10–20 fly heads per genotype were collected with liquid nitrogen and dry ice. Total RNA was extracted using TRIzol reagent following manufacturer's manual (Thermo Fisher Scientific). cDNA was reverse transcribed from 250 or 500 ng of DNase I (M0303S, NEB) treated RNA via MMLV RT (M170A, Promega). A set of five or six standards across 3125-fold dilution was prepared from the equally pooled cDNA of all genotypes in each experiment. Triplicated PCR reactions were prepared in 96-well or 384-well plates for standards and the cDNA sample of each genotype (20- to 40-fold dilution) by mixing in Power SYBR Green Master Mix (Thermo Fisher Scientific) and the following primer sets: ncaqF2: acagagttcacagacgctgag, ncaqR2: ttgctagcgtcaccatatggg; cg7646F: gcctttcgaatgtacgatgtcg, cg7646R: cctagcatgtcataaattgcctgaac or rp49F: cgatatgctaagctgtcgcaca, rp49R: cgcttgttcgatccgtaacc. PCR reactions were performed in Applied Biosystems StepOne (96-wells module) or QuantStudio 6Flex instruments (384 wells module) using standard thermocycle protocols. Melting curve analysis was also performed to evaluate

the quality of the PCR product and avoid contamination. The Ct values were exported as csv files and a standard curve between Ct values and logarithm of dilution was calculated using the liner regression function in Graphpad Prism. The relative expression level for *Nca*, *cg7646* and *rp49* of each sample were estimated by interpolation and anti-logarithm. The expression levels of *Nca* and *cg7646* for each genotype were further normalized to their respective average *rp49* expression level. Statistical differences between the normalized expressions levels of each genotype were determined by Mann-Whitney test or Kruskal-Wallis test with Dunn's post-hoc test using Graphpad Prism.

## Sleep-wake behavioral analysis

Three to five day old male or virgin female flies were collected and loaded into glass tubes containing 4% sucrose and 2% agar (w/v). Sleep-wake behavior was recorded using the *Drosophila* Activity Monitor (DAM, TriKinetics inc MA, USA) system or *Drosophila* ARousal Tracking (DART, BFKlab, UK) in the designated LD regime (12L: 12D, 8L: 16D, DD or LL) at 25°C. Behavioral recordings from the third day of the given LD/DD/LL regime were then analyzed. All flies were entrained to 12L: 12D prior to entering designated LD regimes. For ectopic activation experiments involving UAS-*TrpA1*, flies were cultured in 18°C during development and then entrained to 8L: 16D at 22°C before entering 8L: 16D condition at 27°C. *Drosophila* activity (or wake) is measured by infra-red beam crosses in DAM or by direct movement tracking in DART. A sleep bout is defined by 5 min of inactivity (where inactivity is defined as no beam crosses during 1 min in the DAM or less than 3 mm movement in 5 s in the DART). As a readout of the arousal threshold at ZT4 and ZT16, we measured the proportion of immediate movement initiation in sleeping fly populations (flies that had been immobile for >5 mins before stimulus) upon 5 s of vibration stimuli (five 200 ms 50 Hz pulses with 800 ms intervals) provided by the motors installed within DART system. The csv output files with beam crosses (DAM) or velocity data (DART) were processed by a customized Excel calculators (*Supplementary file 1*) and R-scripts (https://github.com/PatrickKratsch/DAM_analysR) to calculate the following parameters for individual flies: *Onset and offset of each sleep bout*, *sleep bout length*, *day and night sleep minutes*, *daily total sleep minutes*, and *daily sleep profile* (30 min interval).

## Analysis of circadian rhythm strength

An established MATLAB based tool, Flytoolbox, was used for circadian rhythmicity analysis (*Levine et al., 2002a*; *Levine et al., 2002b*). Flies from control and experimental genotypes developed and eclosed under 12L: 12D conditions (25°C). After 3 days of entrainment in 12L: 12D, adult males were transferred into DAM tubes, and circadian rhythmicity of locomotor activity was assessed over eleven days of constant dark (DD) following one initial day of 12L: 12D within the experimental incubator. The strength of rhythmicity (RI) was estimated using the height of the third peak coefficient in the auto-correlogram calculated for the activity time series of each fly. Rhythmic Statistics values were then obtained from the ratio of the RI value to the 95% confidence interval for the correlogram ($2/\sqrt{N}$, where N is the number of observations, which correlatively increase with the sampling frequency), in order to determine statistical significance of any identified period (RS is $\geq$1).

## Immunohistochemistry and confocal microscopy

Adult male flies were anesthetized in 70% ethanol before brains were dissected in PBT (0.1M phosphate buffer with 0.3% Triton-X100) and collected in 4% paraformaldehyde/PBT on ice. The fixation was then performed at room temperature for 15 min before washing 3 times with PBT. The brain samples were blocked using 5% goat serum/PBT for 1 hr at room temperature before incubation with primary antibodies. The samples were washed 6 times with PBT before incubated with Alexa Fluor secondary antibodies in 5% goat serum/PBT at 4°C over 24 hr. After washing 6 times with PBT, the samples were mounted in SlowFade Gold antifade reagent (S36936, Thermo Fisher Scientific) on microscope slides and stored at 4°C until imaged using an inverted confocal microscope Zeiss LSM 710. Primary antibody concentrations were as follows: mouse anti-BRP (nc82, Developmental Studies Hybridoma Bank) - 1:200; rabbit anti-GFP (Invitrogen) - 1:1000; rabbit anti-dsRED (Clontech) - 1:2000. Alexa Fluor secondaries (Invitrogen) were used as follows: Alexa Fluor 647 goat anti-mouse IgG - 1:500, Alexa Fluor 488 goat anti-rabbit IgG - 1:2000, Alexa Fluor 555 goat anti-rabbit IgG - 1:2000. For quantification of nuclei number in *C01 >red* stinger and *A05 >red* stinger brains, unstained Red-Stinger fluorescence was captured via confocal microscopy. DAPI (Sigma Aldrich) was

used to counterstain nuclei (at a dilution of 1:5000). The number of Red-Stinger-positive nuclei in each brain was subsequently quantified using the ImageJ 3D Objects Counter tool, with a variable threshold used to incorporate all of the visible Red-Stinger-positive nuclei. Standardized images from the Virtual Fly Brain can be found in the following files (*Milyaev et al., 2012*; *Manton et al., 2014*):

R14A05-Gal4: http://flybrain.mrc-lmb.cam.ac.uk/vfb/jfrc/fl/reformatted-quant/JFRC2_GMR_14A05_AE_01_13-fA01b_C100226_20100226142935296_02_warp_m0g80c8e1e-1x26r3.nrrd.
R72C01-Gal4: http://flybrain.mrc-lmb.cam.ac.uk/vfb/jfrc/fl/reformatted-quant/JFRC2_GMR_72C01_AE_01_02-fA01b_C091205_20091205104559169_02_warp_m0g80c8e1e-1x26r3.nrrd.
nc82: https://github.com/VirtualFlyBrain/DrosAdultBRAINdomains/blob/master/template/JFRCtemplate2010.nrrd.

## SynaptopHluorin imaging

Synaptic activity of *C01/A05* neurons was monitored in ex vivo fly brains using UAS- super-ecliptic synaptopHluorin construct (UAS-*spH*) (*Miesenböck, 2012*). Adult male *C01/A05* > UAS *spH* or *C01/A05* > UAS *spH, kk* flies were housed in normal behaviour tubes (see behaviour analysis section) and entrained for 3 days in 8L: 16D or LL conditions at 25°C. Individual flies of either genotype were carefully captured between ZT/CT9 and ZT/CT11 and fly brains were immediately dissected in HL3 *Drosophila* saline (70 mM NaCl, 5 mM KCl, 1.5 mM CaCl$_2$, 20 mM MgCl$_2$, 10 mM NaHCO$_3$, 5 mM Trehalose, 115 mM Sucrose and 5 mM HEPES, pH 7.2) at room temperature. Fly brains were transferred into 200 µl HL3 in a poly-lysine treated glass bottom dish (35 mm, 627860, Greiner Bio-One) before imaging using an inverted confocal Zeiss LSM 710 microscope (20x objective with maximum pinhole). Three to five image stacks (12 bits and 16 bits) were taken within two minutes to minimize tissue degradation and to cover the depth of all spH-positive anatomical regions. Z-projections of the image stacks of each brain were generated by ImageJ software before the fluorescent intensity of the indicated neuropil centers was quantified using freely drawn ROIs. Background fluorescence measured by the same ROIs from areas with no brain tissue was then subtracted to obtain the final fluorescent value. Mean fluorescent values of the indicated neuropil regions in each hemisphere were calculated and normalized to the average value of corresponding controls. Medians of the normalized value are compared between genotypes. The statistical difference was determined by Mann-Whitney U-test using Graphpad Prism.

## Bioinformatics

Conservation of amino acid residues between *Drosophila* Neurocalcin and human Hippocalcin was determined using ClustalW2 software for multiple sequence alignment. Amino-acid identity and similarity was visualized using BOXSHADE.

## Acknowledgements

We thank Jason Somers for technical support on infrared camera and DART system installation, Jack Humphrey for performing initial work on *Neurocalcin* knockdown flies, and Kyunghee Koh for helpful comments on the manuscript. This study was supported by the Wellcome Trust (Synaptopathies strategic award [104033]), the MRC [New Investigator Grant MR/P012256/1] and the BBSRC (BB/R02281X/1). PK is supported by a Wellcome Trust Neuroscience PhD studentship (109003/Z/15/Z).

## Additional information

### Funding

| Funder | Grant reference number | Author |
|---|---|---|
| Wellcome | Synaptopathies Strategic Award, 104033 | James Jepson |
| Medical Research Council | MRC New Investigator Award, MR/P012256/1 | James Jepson |

| Wellcome | Graduate Student Fellowship 109003/Z/15/Z | Patrick Krätschmer |
|---|---|---|
| Biotechnology and Biological Sciences Research Council | BB/R02281X/1 | James Jepson |

The funders had no role in study design, data collection and interpretation, or the decision to submit the work for publication.

#### Author contributions
Ko-Fan Chen, Conceptualization, Formal analysis, Validation, Investigation, Visualization, Methodology, Writing—original draft, Writing—review and editing; Simon Lowe, Formal analysis, Validation, Investigation, Visualization, Methodology, Writing—review and editing; Angélique Lamaze, Formal analysis, Investigation, Visualization, Writing—review and editing; Patrick Krätschmer, Software, Formal analysis, Investigation, Visualization, Writing—review and editing; James Jepson, Conceptualization, Supervision, Funding acquisition, Investigation, Writing—original draft, Project administration, Writing—review and editing

#### Author ORCIDs
Ko-Fan Chen (iD) http://orcid.org/0000-0002-7305-6254
James Jepson (iD) http://orcid.org/0000-0002-3357-2801

#### Decision letter and Author response
Decision letter https://doi.org/10.7554/eLife.38114.042
Author response https://doi.org/10.7554/eLife.38114.043

## Additional files

#### Supplementary files
• Supplementary file 1. Customized Excel spreadsheets for sleep calculation.
• Transparent reporting form
DOI: https://doi.org/10.7554/eLife.38114.030

#### Data availability
All data generated or analysed during this study are included in the manuscript and supporting files. Source data files have been provided for all figures and associated supplemental files. Customised R-scripts used to process DAM and DART data are available at https://github.com/PatrickKratsch/DAM_analysR.

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
