## [Decision Letter]

[Editors’ note: a previous version of this study was rejected after peer review, but the authors submitted for reconsideration. The first decision letter after peer review is shown below.]

Thank you for choosing to send your work, "The *Drosophila* dystonia gene homolog *Neurocalcin* facilitates sleep by inhibiting a movement-promoting neural network", for consideration at *eLife*. Your article has been reviewed by three peer reviewers, and the evaluation has been overseen by a Reviewing Editor and a Senior Editor. Although the work is of interest, we regret to inform you that the findings at this stage are too preliminary for further consideration at *eLife*.

Specifically, the reviewers are in agreement that more experiments need to be done in order to elucidate the role of dopamine in the circuit while one reviewer argues that the circuit is really ill defined. We suggest that you carefully read the reviewers’ comments and if you are willing to resubmit this manuscript to *eLife* that you address many of the comments to ensure that the reviewers are satisfied. If you resubmit to *eLife* we will send the manuscript to the same reviewers.

*Reviewer #1:*

The paper is well-constructed and well-written. I'm not sure about its relevance for dystonia, and therefore the overall significance and importance. However, it’s easy to read and follow the logic, and I imagine that the findings also have relevance for people interested in circuits and sleep.

My main comment is to query the importance for dystonia given that there is only a single study linking HPCA to dystonia. The relationship has not been confirmed by multiple groups and/or additional patient cohorts. This should be better explained; some of the dystonia genes mentioned in the Introduction are confirmed as causal, while for others (including HPCA) the causal relationship remains controversial.

The second issue relating to the relevance for dystonia is that it is difficult to know how this dissection of the fly circuitry that utilizes *Nca* relates to mammalian circuity that is relevant for dystonia. Instead, the most important finding (with respect to to dystonia) is the genetic interaction between *Nca* and the dopamine receptor. This is intriguing, however, I don't feel the data establish that the interaction occurs in individual neurons. Instead, it remains possible that KD of the dopamine receptor promotes sleep through a different set of neurons/ circuit – which would represent an indirect interaction rather than the interpretation implied here (genes acting in a common pathway). This issue should be discussed or (better) solved, particularly given the large number neurons/ circuits that affect sleep. Ideally the authors can test whether RNAi KD of Dop1R1 in CO1 and *A05* neurons is sufficient to rescue sleep defects of *Nca* animals. This would provide strong support for a 'direct' molecular relationship between the genes.

A final point is to better explain the relationship between the behavioral assays (that monitor frequency of movement) as a pure read out of animal sleep vs. wake. Is a primary defect in sleep the only possible explanation, or could a different primary defect increase/ cause animal movement in the dark cycle? For example, that *Nca* KD/KO impairs animal ability to detect light/dark, or causes an increased need to feed?

*Reviewer #2:*

In general this appears to be a well done paper. But I have two issues.

1) The genetics is not really up to standard. The authors generated mutants but then only show data from a single homozygosed allele. This is really dangerous. Even "specific" techniques like CRISPR or HR can generate second site mutations. They should show data from a second allele and from transheterozygotes. The fact that the single allele phenocopies RNAi allays a lot of concern, but if you have made the mutants why not use them in a rigorous manner? This is a minor fix to the paper since they report isolating multiple alleles.

2) The circuit analysis is really unsatisfying and a much bigger problem. I am not sure that there is really a "circuit" since the authors have not actually shown any direct connectivity of the implicated cell groups. It is ill-defined and hand-wavy.

I also think there are some other possible interpretations of these results that have not been explicated. One thing that I worry about is that they are just screwing up the brain in some non-specific way. One thing that argues against this (and perhaps bears mention in the paper) is that there are a number of very broad drivers (VGluT, GAD, Tim) that apparently do not have effects- this means that the phenotype is not directly proportional to the number of neurons expressing the RNAi. That is good.

What is less good is that there are other lines that do have phenotype and how those cell groups relate to the A and C lines is not explained. Do they overlap with one or both of these lines? How do they act in combination with these lines? C5GAL4 is a FSB line I think that this neuropil is never mentioned in the context of A and C. Are there multiple "circuits"? Are there hotspots for *Neurocalcin* function? I just am not sure this is very specific.

One thing authors mention is "neurons that innervate the γ-lobes". Are these MBON? Have the authors looked at this? If the gene is required in multiple MBONs this might explain the additivity since MBONs are thought to summate. This should probably be explicitly tested.

I think that until there is a real, connected set of neurons they should not be talking about a circuit. I am not sure that holding the authors to this standard would allow publication of the paper as a revision i.e. there would be too many additional experiments required.

*Reviewer #3:*

This is a very interesting paper that I would be excited to see in print and to recommend to my colleagues – it uses a number of creative approaches to demonstrate a complex, multi-component circuit through which neurocalcin controls sleep. However, because of the broad scope, there are some loose ends that need to be addressed.

First, the importance of dopamine in this circuit has not been demonstrated. Although panel 2G suggests that Dop1R1 is involved in the loss of nighttime sleep, there are no results which directly show that knockdown of Dop1R1 rescues the sleep loss caused by knockdown of neurocalcin specifically in the *A05/C01* circuit implicated in this paper.

Second, Figure 5E and F, which test the prediction that "silencing *C01* and *A05* neurons should suppress sleep loss in *Nca* knockdown in flies", requires some additional controls. The fact that the using *C01/A05* to drive both kk and dOrk results in a significant change from sleep from using these Gal4 drivers to drive dOrk alone suggests that there may be some dilution effects (from having two UAS sequences). To control for this, they need to simultaneously drive knockdown of kk with some other gene with a UAS (such as GFP or synaptophluorin). In addition, they do not show that rescuing neurocalcin in the *A05/C01* circuit alone is sufficient to restore baseline sleep activity.

Lastly, in Figure 2—figure supplement 4C, an n=3 of triplicated qPCR reactions (representing a single biological sample per timepoint) is not sufficient to draw any conclusions. In order to demonstrate that neurocalcin does not cycle, more biological samples are necessary. However, a lack of *Neurocalcin* cycling in the wild type case seems to be tangential to their main point here, that knockdown of *Neurocalcin* does not affect rest activity rhythms. Thus, it may be better to remove the panel altogether.

This paper also focuses extensively on dystonia on a movement disorder, but the data focus primarily on sleep – while the connection between the two is well explained in the last paragraph of the Discussion, revising the Introduction, the description of the results (subsection “NCA promotes sleep by suppressing synaptic output from a wake -promoting circuit”, second paragraph, for instance), and the first paragraph of the Introduction will be helpful for the reader. Alternately, describing locomotor activity in addition to sleep in Figures 2 through 5 (through measurements such as speed and activity counts per minute) will help emphasize locomotion, rather than sleep. In Figure 1—figure supplement 2, activity counts per waking minute should be reported, not total beam breaks.

[Editors’ note: what now follows is the decision letter after the authors submitted for further consideration.]

Thank you for resubmitting your article "*Neurocalcin* regulates nighttime sleep and arousal in *Drosophila*" for consideration by *eLife*. Your article has been reviewed by three peer reviewers, one of whom, Hugo Bellen, is a member of our Board of Reviewing Editors, and the evaluation has been overseen by Vijay Raghavan as the Senior Editor.

The reviewers have discussed the reviews with one another and the Reviewing Editor has drafted this decision to help you prepare a revised submission.

This manuscript identifies a novel function for the gene, Neurocalcin (*Nca*) in *Drosophila*. The data presented support a role for *Nca* in specifically promoting nighttime sleep and identify some neurons, predominantly in the mushroom bodies, which are required for this phenotype.

Summary:

It has been previously noted that dystonia in humans and sleep in *Drosophila* share a common molecular pathway. Therefore in an attempt to find new genes that regulate sleep in *Drosophila*, the authors began by looking for sleep defects in flies after disrupting the fly homolog of human dystonia associated genes. This led to the identification of *Nca*, as pan-neuronal RNAi knockdown of *Nca* caused a significant decrease in the amount of nighttime sleep. This phenotype was confirmed with the use of multiple CRISPR generated alleles which knockout the entire *Nca* coding sequence. Furthermore, *Nca*'s control of sleep/arousal during the 'night' is dependent both on the circadian clock and the ability to sense light. In an attempt to identify the specific neurons that require *Nca* to regulate sleep, they used RNAi to screen a number of GAL4 drivers that expresses in specific neuronal subsets looking for drivers that were able to phenocopy the sleep defect seen with pan-neuronal knockdown. While they did not find a single driver that could phenocopy, they did find that co-expression with *C01* and *A05* drivers (both enhancer GAL4 lines) was able to cause a deficit in nighttime sleep. Both of these drivers are expressed in a subset of mushroom body neurons. Using Synapto-pHluorin as a measure of synaptic release suggested that there was less synaptic release in the AMMC and α/β lobes of the mushroom body with RNAi knockdown of *Nca*. In fact, excitation of the *C01* neurons, using TrpA1, alone can promote wakefulness while silencing both *C01* and *A05* neurons is necessary to suppress the loss of nighttime sleep with *Nca* KD. Together their model suggests that *Nca* regulates synaptic activity in a subset of mushroom body neurons to regulate sleep/arousal.

Overall, the manuscript is well written and interesting and will expand our understanding of sleep regulation. The authors have done a good job in addressing the concerns of the reviewers from the initial evaluation. However, many of the conclusions have changed from the original manuscript and therefore, we have a number of suggestions to improve this version of the manuscript.

Essential Revisions:

1) The story would benefit from a more mechanistic explanation, especially linking the data in Figure 3 to the data in Figure 6. One such experiment would be to manipulate the *A05, C01*, and OK107 knockdowns of neurocalcin (and relevant combinations) in the arousal assay, in DD, and in LL, similar to what was done in Figure 2 and Figure 3A and G for the pan-neuronal knockdown, and possibly combining the LL treatment with some of the functional imaging presented in Figure 5.

2) While doing the experiments described in (1) would make the story more comprehensive, more detail may be necessary to link the findings related to cryptochrome to the neuronal populations described in Figure 6. Is neurocalcin's response to light also happening within the cryptochrome positive neurons, or are the responses of cryptochrome positive neurons interacting with the populations described in Figure 6 to regulate sleep?

3) The explanation for Figure 3 is somewhat confusing. Do control flies in the LL case exhibit similar quantities of sleep to neurocalcin knockdown because light inhibits neurocalcin function in the wild type case? This seems unlikely since in the presence of light, the controls do not show a reduction in sleep. The fact that the flies sleep more in constant light than in constant dark also seems extremely unusual.

4) Which neurons express *Nca* protein, is it really expressed by all neurons as the authors indicate is previously shown? Teng et al., 1994, which is the only report of *Nca* expression in the fly, simply states that *Nca* is expressed "throughout the central nervous system" but only looked at adult brain slices with images that are too low quality to make any claim that all neurons express this protein. Furthermore, do *Nca* protein levels vary between sleep-wake states to explain why it is only important during sleep? Rat and likely other mammalian Hippocalcin antibodies are commercially available, given that the Teng et al. paper successfully used a rat antibody to label fly brains, it would be worth trying one or more of these antibodies both for immunohistochemistry to determine expression pattern and western blots to assess if *Nca* cycles circadian or in response to light/dark.

5) How does *Nca* inhibit neuronal activity in the context of sleep/wake or light/dark? The previous manuscript suggested it was dopamine receptors, now the Discussion mentions NMDA receptor internalization. Can this be tested genetically and/or by looking at protein localization?

6) The Results and Discussion points about the distinct roles of *A05* and *C01* (pro-arousal and modulatory, respectively) feel over-interpreted.

[Editors' note: further revisions were requested prior to acceptance, as described below.]

Thank you for resubmitting your work entitled "Neurocalcin regulates nighttime sleep and arousal in *Drosophila*" for further consideration at eLife. Your revised article has been favorably evaluated by K VijayRaghavan as the Senior Editor, a Reviewing Editor, and three reviewers.

The manuscript has been improved but there are some remaining issues that need to be addressed before acceptance, as outlined below. Please address these to the best of your ability.

1) Figure 4H: was the axis label supposed to read "percentage of flies responding to night time stimulus" (rather than daytime stimulus)?

2) There was a typo in an earlier review when it was requested the authors report the combinations of Gal4 drivers that did not yield negative results. This should have asked what combinations of Gal4 drivers the authors did test, since this kind of negative data is useful for interpreting how exclusive the network of neurons involved is. As it stands, the reason why the *C01* and *A05* Gal4 drivers were chosen as the specific combinations seems a little odd.

3) Results section, fifth paragraph: "To test whether sleep loss caused by neuronal *Nca* knockdown flies was due to [...]"

The authors ask if the clock is affected by NCA KD and show constant dark records. Under what light conditions were those flies entrained? It would be useful to also show the activity patterns under LD conditions, preceding release into DD.

4) The authors test a model of NCA action in *A05* and *C01* neurons in two ways. First with TrpA1 activation. Second with *dORK* silencing. I find the first experiment reasonable, but the second experiment much less compelling. The model postulates that *A05/C01* neuron activity is regulated by NCA and that more activity by those neurons promotes waking and disrupts night time sleep. However, constitutive *dORK* expression is known to irreversibly damage neurons. The *dORK* over-expression experiment shows that the loss of sleep in the *Nca* KD requires *A05/C01* neurons, but does not distinguish between suppression of neuronal activity, or a non-specific decline in neuron health. As I understand it, either outcome could produce the observed result.

5) "Instead, sleep-relevant NCA activity can largely be localized to two distinct domains of the Drosophila nervous system defined by the *A05*- and *C01*-Gal4 drivers"

This conclusion is based on spatially limiting the RNAi effects, thus defining brain regions that are necessary. It does not preclude involvement of other regions. Therefore this conclusion over-interprets the data.

6) Figure 1—figure supplements 2 & 6: the cartoon representations of *Nca* and its immediate neighbor CG7646 are confusing. In supplement 2, *Nca* and CG7646 appear to share two 5'UT exons in common. In supplement 6, CG7646 appears to be distinct and non-overlapping. Which is true? Did the authors test the sleep/activity behaviors of flies with CG7646 KD by RNAi?

[Editors' note: further revisions were requested prior to acceptance, as described below.]

Thank you for resubmitting your work entitled "Neurocalcin regulates nighttime sleep and arousal in *Drosophila*" for further consideration at eLife. Your revised article has been favorably evaluated by K VijayRaghavan (Senior Editor), a Reviewing Editor, and one reviewer.

The manuscript has been improved but there are still some remaining issues that need to be addressed with before acceptance. Please make textual changes according to the suggestions of the reviewer as outlined below:

The authors have addressed all the points raised in the last round of review. For the most part, I feel the responses were appropriate. In some cases, I have continued questions.

Interpretation of *dORK* effects - The authors make a forceful argument against the possibility of prolonged *dORK* mis-expression causing irreversible developmental effects. They suggest that the lethality observed with driving hid or reaper transgenes contrasts with the viability of driving *dORK* and hence the neurons must be "healthy". I disagree because such lethality may well result from damage to non-neuronal tissues: the fly-light lines are described for their neural expression patterns, but non-neuronal expression is not generally described or precluded. Hence this result does not bear on the question of neuronal health following prolonged *dORK* expression. Likewise, it is not clear why locomotor activity levels in flies would necessarily be affected if *C01*- and *A05* neurons were partially damaged.

The use of UAS-shi[ts] in the *Nca* knockdown background is more compelling because it is a conditional design and because it provides a result in line with the hypothesis. However, it is not included in the paper for technical reasons so it cannot stand as a "complete" result, and likewise it cannot serve as a sub-textual proxy to justify inclusion and interpretation of the prolonged *dORK* expression experiment.

I was wrong in stating the literature documents irreversible damage to *Drosophila* neurons with prolonged *dORK* expression and so I agree with the authors in their statement to that regard. However, the literature is clear in documenting the irreversible damage to *Drosophila* neurons with prolonged Kir expression, which as shown by White and colleagues, works in comparable fashion to *dORK* by imposing a hyperpolarized state. I suggest the important issue is whether prolonged hyperpolarization of *Drosophila* neurons is (beyond reasonable doubt) an effective means to exclusively affect neurotransmission. Ceriani and colleagues in 2011 showed that prolonged expression of Kir in LNv pacemaker neurons irreversibly affected molecular oscillations, structural maturation and neuropeptide content. In contrast, restricting Kir expression to adult stages, led to outcomes that were at least partially reversible. Therefore, I hold to the opinion that the present *dORK* experiment is not compelling, in spite of the additional observations provided. If the authors insist on its inclusion, then I think it's important to share with readers a clear caveat based on the Ceriani work.

Consideration of CG7646 - I appreciate the clarification of the complicated genomic arrangement of CG7646 and *Nca*. It is helpful to know they share 5'UTR sequences but different CDS exons. I think it would be useful to make this point clear in the text - for example on line 111 of the revision, CG7646 is described as a "neighboring" locus. I interpret "neighbor" as independent or free-standing. These two genes are overlapping or in some fashion "compound". I recommend the text include a clear description of this genomic arrangement and be re-examined to ensure clarity on this point in all mentions. Likewise, I suggest inclusion of the RNAi knockdown results of CG7646 as supplemental information to allay concerns on this point. Regarding that experiment, one other minor point: in the response and author response images, the authors describe using two different RNAi's from NIG. However, my reading of the NIG site indicates availability of only one RNAi for CG7646, albeit offered as a II and III chromosome insertion. The figure leads the reader to think two different RNAis were tested. The text should be clear as to whether the same RNAi at different insertion sites, or different RNAi's were tested.

---

## [Author Response]

[Editors’ note: the author responses to the first round of peer review follow.]

Specifically, the reviewers are in agreement that more experiments need to be done in order to elucidate the role of dopamine in the circuit while one reviewer argues that the circuit is really ill defined. We suggest that you carefully read the reviewers’ comments and if you are willing to resubmit this manuscript to eLife that you address many of the comments to ensure that the reviewers are satisfied. If you resubmit to eLife we will send the manuscript to the same reviewers.

We sincerely thank the reviewers for their insightful comments relating to our manuscript. Collectively, these have prompted us to strengthen the focus of the paper more appropriately towards the role of *Neurocalcin* as a novel sleep gene in *Drosophila*. As noted by reviewer 1, current data suggest that Hippocalcin mutations are rare amongst dystonia patients. In our revised manuscript, we therefore place less emphasis on the potential role of the human *Neurocalcin* homolog Hippocalcin/*HPCA* in dystonia. Correspondingly, we have altered our title to underscore this change of focus, which now reads: ‘Neurocalcin regulates nighttime sleep and arousal in *Drosophila*’.

Nonetheless, the potential genetic links between human dystonia and *Drosophila* sleep were in fact highly relevant to how we initially identified *Neurocalcin* as a sleep gene. For brevity, we initially omitted the actual screening strategy that led us to focus on *Neurocalcin*, but we now describe this in more detail in our Introduction as we believe this approach will be of interest to the community.

Importantly, our revised manuscript not only addresses the reviewer concerns detailed below but also contains abundant new mechanistic data regarding how Neurocalcin solely promotes night sleep.

Firstly, in our revised Figure 2, we combine video tracking with mechanical stimulation to probe daytime versus night time arousal in *Neurocalcin* knockdown, knockout and respective control flies. We find that knockdown or knockout of *Neurocalcin* does not impact the probability of response to a mechanical stimulus during the day (Figure 2A-B, E), but significantly lowers the arousal threshold during the night (Figure 2C-D, F). Arousal thresholds are known to dynamically vary over 24 h in *Drosophila*, with day sleep having a lower arousal threshold than night sleep (Faville et al., 2015; van Alphen et al., 2013). However, known genetic regulators of this day/night difference in arousal are scarce. Our identification of Neurocalcin as a specific regulator of night time arousal will therefore be of substantial interest to the sleep field.

Secondly, we have investigated how this night-specific effect is gated. We show in our revised Figure 3 that both the circadian clock and light are capable of gating *when* Neurocalcin promotes sleep, likely via inhibiting the wake-promoting effect of loss of Neurocalcin during the day. We also present data showing that the CRY blue-light photoreceptor is involved in this process.

Collectively, these new data combined with alterations to the manuscript significantly enhance both our mechanistic understanding of how Neurocalcin impacts night sleep and the overall robustness of our work. Below, we further respond to each individual reviewer comment.

Reviewer #1:The paper is well-constructed and well-written. I'm not sure about its relevance for dystonia, and therefore the overall significance and importance. However, it’s easy to read and follow the logic, and I imagine that the findings also have relevance for people interested in circuits and sleep.My main comment is to query the importance for dystonia given that there is only a single study linking HPCA to dystonia. The relationship has not been confirmed by multiple groups and/or additional patient cohorts. This should be better explained; some of the dystonia genes mentioned in the Introduction are confirmed as causal, while for others (including HPCA) the causal relationship remains controversial.

We fully agree with the reviewer. We have therefore made the rare nature of *HPCA* mutations in dystonia clear in both the Introduction (subsection “Identification of *Neurocalcin* as a sleep-promoting factor”, second paragraph) and in the Discussion (last paragraph), alongside appropriate references.

The second issue relating to the relevance for dystonia is that it is difficult to know how this dissection of the fly circuitry that utilizes Nca relates to mammalian circuity that is relevant for dystonia. Instead, the most important finding (wrt to dystonia) is the genetic interaction between Nca and the dopamine receptor. This is intriguing, however, I don't feel the data establish that the interaction occurs in individual neurons. Instead, it remains possible that KD of the dopamine receptor promotes sleep through a different set of neurons/ circuit – which would represent an indirect interaction rather than the interpretation implied here (genes acting in a common pathway). This issue should be discussed or (better) solved, particularly given the large number neurons/ circuits that affect sleep. Ideally the authors can test whether RNAi KD of Dop1R1 in CO1 and A05 neurons is sufficient to rescue sleep defects of Nca animals. This would provide strong support for a 'direct' molecular relationship between the genes.

We performed a series of experiments to address this important question. However, before describing these, it is relevant to give a brief overview of the particular mutant lines used to examine epistatic interactions between *Neurocalcin* and *Dop1R1.*

In our initial experiments we used two transposon insertions in the *Dop1R1* locus (on chromosome III of *Drosophila*) as independent alleles that were likely loss-of-function. Due to the known importance of genetic background on sleep in *Drosophila* (Cirelli et al., 2005), we swapped chromosomes I and II from these mutant lines with corresponding chromosomes derived from our isogenic iso31 control strain. However, we could not outcross chromosome III itself while easily following the *Dop1R1* alleles themselves, as there was no visible marker within the transposons that could be observed in the iso31 background.

To both better control for genetic background and achieve cell-specific *Dop1R1* knockdown, we therefore used two complementary approaches. Firstly, we outcrossed a transgenic RNAi line targeting *Dop1R1* for 5 generations into the iso31 background. Secondly, we outcrossed a previously characterised hypomorphic allele of *Dop1R1* called *dumb*^2^ (Kim et al., 2007), also for 5 generations. However, in contrast to our previous results where heterozygosity for either *Dop1R1* allele rescued sleep loss in *Neurocalcin* knockdown flies, neither expression of outcrossed *Dop1R1* RNAi, nor homozygosity for the outcrossed *dumb*^2^ allele, suppressed the effect of pan-neuronal *Neurocalcin* knockdown (Author response image 1; original results are shown in Author response image 1 for reference).

**Author response image 1. respfig1:** Sleep loss in *Nca*^KO^ or *Nca*^KD^ flies is not modified by altered expression of *Dop1R1*. (A-B) Night sleep levels in *Nca*^KD^ (*elav* > *kk*) adult males co-expressing control UAS*GFP* RNAi (GFP, dashed red lines) are significantly reduced compared to controls (grey lines).However, this profile does not differ from flies co-expressing UAS-*Dop1R1* and *kk* (*Nca*) RNAi (D1, purple lines). Mean sleep levels are shown in (**A**) across 24 h in L8: D16 conditions, while median night sleep levels are shown in (**B**). n = 16-24. (C-D) Night sleep levels in *Nca*^KO^ (red lines) flies are reduced as compared to *iso31* controls (grey line). However, this sleep loss is not suppressed by homozygosity for the hypomorphic *dumb*2 allele of *Dop1R1* (purple line). n = 19-24. (E-F) heterozygosity of the non-isogenized hypomorphic *Dop1R1* allele *Dop1R1*MI03085-GFST.2(*D1mic*/+) suppressed sleep loss in *Nca*^KD^ adult males. Mean sleep patterns were shown in the original Figure 2G. Median total night sleep levels were shown in the original Figure 2H. n = 15-48. *p < 0.05, ** p <0.01, *** p <0.001, ns– p > 0.05, Kruskal-Wallis test with Dunn’s post-hoc test. Control and experimental genotypes are indicated by color scheme and black dots.

Given these conflicting results, which we assume are due to differences in genetic background between the fully and partially outcrossed Dop1R1 alleles and RNAi insertions, we have removed Figure 2G, H from the manuscript. Importantly, despite this removal the overall mechanistic strength of our paper has been strongly enhanced due to our new analyses of how Neurocalcin differentially regulates day/night arousal thresholds and how circadian/light-sensing pathways in turn gate when Neurocalcin impacts sleep, detailed above and in the revised manuscript as Figures 2 and 3.

A final point is to better explain the relationship between the behavioral assays (that monitor frequency of movement) as a pure read out of animal sleep vs. wake. Is a primary defect in sleep the only possible explanation, or could a different primary defect increase/ cause animal movement in the dark cycle? For example, that Nca KD/KO impairs animal ability to detect light/dark, or causes an increased need to feed?

Several independent lines of evidence argue for a role for *Neurocalcin* in regulating sleep rather than an independent physiological variable that could cause nighttime hyperactivity.

Firstly, our new data analysing arousal thresholds in *Neurocalcin* knockdown and knockout flies (Figure 2 in our revised manuscript) strongly argue that *Neurocalcin* is acting to specifically modulate the arousal threshold during the night, providing a mechanistic basis by which *Neurocalcin* impacts sleep (described in the subsection “NCA suppresses nighttime arousal”).

Secondly, the fact that sleep loss in *Neurocalcin* knockdown flies only occurs in the subjective night of constant dark conditions (new data presented in Figure 3A in our revised manuscript) shows that loss of *Neurocalcin* does not simply cause hyperactivity in the absence of light. Instead, sleep loss is clearly regulated by the circadian clock. This again supports a direct sleep-regulatory role of *Neurocalcin*, since sleep onset is gated by the circadian clock (Borbely, 1982). We have made this point clear in the first paragraph of the subsection “Light-sensing and circadian pathways define when NCA promotes sleep”.

Finally, the clear and sustained startle-responses to lights-off in *Neurocalcin* knockout males (Figure 1N, O of revised manuscript) also strongly suggests that these mutants are able to detect the absence of light.

We cannot fully rule out the possibility that *Neurocalcin* regulates hunger specifically during the night, but this hunger would have to be gated by the clock and light in such a way that it was suppressed during constant light conditions, as we observe no sleep loss in *Neurocalcin* knockdown flies under this condition (see Figure 3G, H in revised text). We consider it far more likely that *Neurocalcin* is indeed a bona fide sleep-promoting gene.

Reviewer #2:In general this appears to be a well done paper. But I have two issues.1) The genetics is not really up to standard. The authors generated mutants but then only show data from a single homozygosed allele. This is really dangerous. Even "specific" techniques like CRISPR or HR can generate second site mutations. They should show data from a second allele and from transheterozygotes. The fact that the single allele phenocopies RNAi allays a lot of concern, but if you have made the mutants why not use them in a rigorous manner? This is a minor fix to the paper since they report isolating multiple alleles.

To address this point we have now measured sleep levels in three additional combinations of *Nca* knockout alleles (Figure 1—figure supplement 6). Homozygosity for the *Nca*^KO2^ allele or trans-heterozygous combinations between the *Nca*^KO1^ allele and either *Nca*^KO2^ or *Nca*^KO3^ all exhibited significant reductions in night sleep (Figure 1—figure supplement 6A, B). Thus, in total we have shown using *four* combinations of *Nca* knockout alleles and *three* independent *Nca* RNAi lines that reducing NCA expression consistently results in night-specific sleep loss. Collectively, these results robustly support a role for NCA in regulating night sleep.

2) The circuit analysis is really unsatisfying and a much bigger problem. I am not sure that there is really a "circuit" since the authors have not actually shown any direct connectivity of the implicated cell groups. It is ill-defined and hand-wavy.I also think there are some other possible interpretations of these results that have not been explicated. One thing that I worry about is that they are just screwing up the brain in some non-specific way. One thing that argues against this (and perhaps bears mention in the paper) is that there are a number of very broad drivers (VGluT, GAD, Tim) that apparently do not have effects- this means that the phenotype is not directly proportional to the number of neurons expressing the RNAi. That is good.

We agree with the reviewer that *Neurocalcin* knockdown in several broad neuronal subsets argues against a generalised neuronal dysfunction, and have made this point more explicit in the revised text (subsection “NCA acts in two neuronal subpopulations to promote night sleep”, third paragraph). The fact that we have narrowed down *Neurocalcin’s* sleep-relevant activity to approximately 350 neurons argues against a generalised ‘neuropathy-like’ effect of *Neurocalcin* knockdown/knockout.

What is less good is that there are other lines that do have phenotype and how those cell groups relate to the A and C lines is not explained. Do they overlap with one or both of these lines? How do they act in combination with these lines? C5GAL4 is a FSB line I think that this neuropil is never mentioned in the context of A and C. Are there multiple "circuits"? Are there hotspots for Neurocalcin function? I just am not sure this is very specific.

To clarify, in our mini-screen of individual neuronal subpopulations, there was no subset in which *Neurocalcin* knockdown resulted in night sleep loss, including *C5*Gal4 and two other FSB-positive driver lines (see Figure 4—figure supplement 1 of revised manuscript). We believe the colour scheme in our original figure may have been slightly confusing, so we have changed the colour-coding and the text within the figure accordingly to emphasise that *only Neurocalcin* RNAi driven by *elav*-, *nsyb*- and *inc*-Gal4 yielded significant sleep loss relative to both driver and transgene alone controls.

Although we have narrowed the number of neurons in which *Neurocalcin* acts to impact sleep from > 100000 to approximately 350, it is clear from our results that *Neurocalcin* is required in multiple neuronal regions. This is consistent with many studies of sleep genes in *Drosophila*, including *sleepless*, *insomniac*, *taranis* and *cyclin-A*, all of which failed to identify a single neuropil region that fully accounted for sleep loss in these mutants. We note that previously studied sleep regions such as the fan-shaped body (Donlea et al., 2011), large LN_v_ neurons (Parisky et al., 2008; Shang et al., 2008), and ellipsoid body (Liu et al., 2016), are not present in either *C01*- and *A05-*neurons, suggesting novel sleep-regulatory circuits are present within these expression domains. Identifying these subdomains will be a fruitful avenue of future investigation.

One thing authors mention is "neurons that innervate the γ-lobes". Are these MBON? Have the authors looked at this? If the gene is required in multiple MBONs this might explain the additivity since MBONs are thought to summate. This should probably be explicitly tested.

We agree with the reviewer that this is an interesting question. To properly clarify the involvement of MBONs in *Nca*^KD^ mediated sleep loss, a detailed investigation of the overlap between MBONs (Aso et al., 2014) and *R14A05*- (*A05-*) or *R72C01-*positive (*C01*-) neurons via orthogonal labelling will be required. However, this approach is currently unavailable to us due to the lack of *A05*-LexA or *C01*-LexA driver lines that could be combined with MBON-Gal4 lines.

Nonetheless, since we observed no clear difference between our own images of *C01*- and *A05*-Gal4 expression within the MB region (Figure 4A, C) and those acquired by the Janelia Flylight team (Jenett et al., 2012), we used the standardised published images available from Virtual Fly Brain (www.virtualflybrain.org) to digitally examine the overlap between *C01*- and *A05*-Gal4 in the MB region (Figure 5—figure supplement 2A).

Using this approach, it is clear that *A05*- and *C01*-Gal4 label the α’β’ and αβ lobes of MB intrinsic Kenyon cells respectively (Figure 5—figure supplement 2A, Author response image 2). We also observed potential labelling of β’1γ3 MBONs by *C01*-Gal4, while β’1-2 and γ1-2 MBONs may be labelled by *A05*-Gal4. Finally, despite lacking their characteristic tiling pattern (Aso et al., 2014), we cannot rule out the possibility that *A05*- and *C01*-Gal4 express in the α1-3 or α’1-3 MBONs (Aso et al., 2014).

In our original manuscript, we combined the pan-MB driver *ok107*-Gal4 with *A05*Gal4 to show that *Nca* knockdown in all MB KCs and potentially components of α’β’ and γ1-2 MBONs neurons (*A05/ok107* > *kk*) causes modest but significant night sleep loss (shown in Author response image 2). We have now combined *ok107*-Gal4 with *C01*-Gal4 to show that *Nca* knockdown in all MB KCs potentially alongside β’1 γ3 MBONs (*C01*/*ok107* > kk) has no impact on night sleep (Figure 5—figure supplement 2G). For comparisons of effect sizes, the reduction in night sleep caused by *Nca* knockdown in *A05*- and *C01*-neurons is shown in Author response image 2. These findings suggest that MB-KCs as well as parts of the α’1-3, β2’ and γ1-2 MBONs may modestly contribute to *Nca*-mediated sleep, while the β’1γ3 MBONs are likely not involved.

We have set out to further examine this hypothesis as a separate follow-up project by using *A05*-Gal4 in parallel with an array of defined MBON split-Gal4 drivers. Initially, we reduced *Nca* levels in β’1γ3, γ2α’1 and α2p MBONs alone. These manipulations did not result in night sleep loss (Author response image 2). We then combined the *A05*Gal4 driver with one of 3 defined MBON drivers. We found that only simultaneous *Nca* knockdown using *A05*-Gal4 and the α2p MBON split-Gal4 driver resulted in significant night sleep loss (Author response image 2).

**Author response image 2. respfig2:** Modestcontribution of MB output neurons (MBONs) to NCA-mediated sleep. (**A**) Confocal stacks labelling *R14A05*- (*A05*, green) and *R72C01*- (*C01*, magenta) positive neurons. Images are from (Jenett et al., 2012) and deposited at Virtual Fly Brain (www.virtualflybrain.org). Images were downloaded and digitally superimposed (Merged) onto a standardised fly brain (active zones are labelled in blue using the nc82 antibody; the image source data is distributed under a CC BY-NC-SA 4.0 license). (**B**) Tiling scheme of MBONs adapted from (Aso et al., 2014). The *A05*- (green) and *C01*-Gal4 (magenta) drivers potentially label complementary parts of MBONs (blue letters), though there is potential limited overlap within the β’1 region. *C01*-Gal4 clearly labels the αβ lobe of MB intrinsic Kenyon cells αβ-KC, black letters), while *A05*-Gal4 labels α’β’-KC. (**C**) Simultaneous *Nca* knockdown in *A05*-neurons (potentially including α’β’-KC and α’β’-MBONs) and all KC neurons (using *ok107*Gal4) results in limited night sleep loss. (**D**) Simultaneous *Nca* knockdown in *C01*-neurons (including αβ-KC and γ3-MBONs) and all KC neurons (using *ok107*-Gal4) did not result in significant night sleep loss. (**E**) *Nca* knockdown in both *A05*- and *C01*-positive neurons results in robust night sleep loss. For (C-E), please see Figure 5—figure supplement 2G for nvalues. (**F**) Knockdown of *Nca* in *A05*-neurons (which potentially include α’β’-KCs and α’β’MBONs) and in 3 distinct MBON regions potentially labelled by *A05*-Gal4 (γ2α’1) or *C01*-Gal4 (β’1γ3 and α2p) does not cause sleep loss. Simultaneous *Nca* knockdown in both *A05*-neurons and in α2p MBONs results in modest sleep loss (*A05*/*MB542B* > kk, red). In contrast, no sleep reduction was observed following *Nca* knockdown in *A05* (β’1γ3 MBON or *A05*/γ2α’1 MBONs. n = 16-28. *p < 0.05, **p < 0.01, ***p < 0.001, ns – p > 0.05, compared to driver and RNAi alone controls by Kruskal-Wallis test with Dunn’s post-hoc test.

These initial results are in concordance with the reviewer’s proposition and indicate that the αβ- and α’β’- KCs, and perhaps their downstream β’2 and α2 MBONs, may additively contribute to sleep loss in *Nca* knockdown flies. Nonetheless, it is important to note that the contribution of these neurons is modest.

While these results have provided us with potential insights into the contribution of subsets of MBONs to sleep loss in *Nca* knockdown flies, a comprehensive *Nca* knockdown screen with higher resolution within KCs and MBONs without involving *A05*- and *C01*-Gal4 will be required in the long term. Moreover, we interpret this data with caution because the MBON split-Gal4 driver lines have not been outcrossed into the iso31 background. Since the effect of *Nca* knockdown on sleep appears to be susceptible to modification by genetic background (Author response image 1), we will solely include data from the outcrossed *A05*-, *C01*-, and *ok107*-Gal4 lines in our manuscript, and follow up these studies once the individual components of the MBON split-Gal4 lines are fully outcrossed and subsequently recombined.

I think that until there is a real, connected set of neurons they should not be talking about a circuit. I am not sure that holding the authors to this standard would allow publication of the paper as a revision i.e. there would be too many additional experiments required.

We concur with the reviewer. We have therefore removed explicit mention of ‘circuits’ in the text, which might imply that the *C01*- and *A05*-domains are functionally connected. We have instead emphasised that *Neurocalcin* is required in a dispersed network consisting of multiple neuropil domains (see Discussion, second paragraph). This conclusion is more strongly supported by our *Neurocalcin* knockdown data presented in Figure 4 and associated figure supplements 1-4.

Reviewer #3:This is a very interesting paper that I would be excited to see in print and to recommend to my colleagues – it uses a number of creative approaches to demonstrate a complex, multi-component circuit through which neurocalcin controls sleep. However, because of the broad scope, there are some loose ends that need to be addressed.First, the importance of dopamine in this circuit has not been demonstrated. Although panel 2G suggests that Dop1R1 is involved in the loss of nighttime sleep, there are no results which directly show that knockdown of Dop1R1 rescues the sleep loss caused by knockdown of neurocalcin specifically in the A05/C01 circuit implicated in this paper.

We attempted to perform cell-specific knockdown experiments to examine this interesting question. However, as described above and in Author response image 1, due to inconsistencies between different *Dop1R1* alleles and RNAi lines (likely due to differences in genetic background) we have decided to remove the data shown originally in Figure 2 describing a genetic interaction between *Neurocalcin* and *Dop1R1*. To compensate, we now add substantial *mechanistic* data revealing a specific role for Neurocalcin in regulating arousal during the night but not the day, and illustrating a dual role for circadian and light-sensing pathways in gating the timing of this effect (presented as Figures 2 and 3 in revised text).

Second, Figure 5E and F, which test the prediction that "silencing C01 and A05 neurons should suppress sleep loss in Nca knockdown in flies", requires some additional controls. The fact that the using C01/A05 to drive both kk and dOrk results in a significant change from sleep from using these Gal4 drivers to drive dOrk alone suggests that there may be some dilution effects (from having two UAS sequences). To control for this, they need to simultaneously drive knockdown of kk with some other gene with a UAS (such as GFP or synaptophluorin).

We have repeated the silencing experiment using UAS-FRT-stop-FRTCD8::GFP as an irrelevant control transgene expressed alongside *Neurocalcin* RNAi (Figure 8E, F). Importantly, co-expression of *dOrk* suppresses sleep loss in *Neurocalcin* knockdown flies when compared to flies expressing both *Neurocalcin* RNAi and UASFRT-stop-FRT-CD8::GFP.

In addition, they do not show that rescuing neurocalcin in the A05/C01 circuit alone is sufficient to restore baseline sleep activity.

We attempted to address this point. We first generated a UAS-*Nca* transgenic fly line using a clone derived from BDGP Tagged ORF Collection (UFO04182). Surprisingly, expression of this transgene failed to rescue *Nca* knockout flies. We therefore made V5-tagged or untagged UAS-*Nca* transgenes in-house and generated a second set of stable transgenic fly lines. These also failed to rescue the *Nca* knockout or knockdown phenotypes when expressed in neurons using the *elav*-Gal4 driver.

We believe we now understand the reason for this lack of rescue. The *Nca* sequence within these transgenes contained both 5’ and 3 UTR sequences as documented in the *Drosophila* genome browser (http://flybase.org/cgi-bin/gbrowse2/dmel/?Search=1;name=FBgn0013303) (FB2018_02). However, there is a conserved element upstream of the documented 5’ UTR that was not included within the transgene sequence. Our recent data suggests that the lack of this sequence may be drastically limiting expression of transgenic NCA in neurons, thus limiting the ability to rescue the mutant/knockdown phenotypes.

**Author response image 3. respfig3:** 

Confocal images showing adult male brain expressing either *CD8::GFP* (A) or UAS-*Nca*-V5 (B) under control of the *R72C01*-Gal4 (*C01*) driver. A negative control for leaky transgene expression (transgene insertion alone in the absence of the *C01* driver) is shown in (C).

In Author response image 3, we show either UAS-*CD8::GFP* or UAS-*Nca*-V5 expressed via the same Gal4 driver, *C01*-Gal4. *C01*-driven CD8::GFP can be observed in multiple neuropil regions, including the AMMC, mushroom bodies and many isolated cell bodies (Author response image 7). In contrast, *C01*-driven NCA-V5 could only be observed in the AMMC (Author response image 7). This staining was specific as an anti-V5 antibody alone did not label any neuropil region (Author response image 7). Clearly, there is an unknown element lacking in our transgene sequence that is limiting *Neurocalcin* protein expression, which we are now attempting to identify.

Unfortunately, these complications have precluded addition of rescue data in our current manuscript. Nonetheless, it is important to note that *four* combinations of independent *Neurocalcin* knockout alleles (see Figure 1—figure supplement 6) and *three* independent RNAi lines all yield consistent night-specific sleep phenotypes. Thus, the link between *Neurocalcin* and night sleep is genetically robust.

Lastly, in Figure 2—figure supplement 4C, an n=3 of triplicated qPCR reactions (representing a single biological sample per timepoint) is not sufficient to draw any conclusions. In order to demonstrate that neurocalcin does not cycle, more biological samples are necessary. However, a lack of Neurocalcin cycling in the wild type case seems to be tangential to their main point here, that knockdown of Neurocalcin does not affect rest activity rhythms. Thus, it may be better to remove the panel altogether.

We have removed this panel from the manuscript.

This paper also focuses extensively on dystonia on a movement disorder, but the data focus primarily on sleep – while the connection between the two is well explained in the last paragraph of the Discussion, revising the Introduction, the description of the results (subsection “NCA promotes sleep by suppressing synaptic output from a wake -promoting circuit”, second paragraph, for instance), and the first paragraph of the Introduction will be helpful for the reader. Alternately, describing locomotor activity in addition to sleep in Figures 2 through 5 (through measurements such as speed and activity counts per minute) will help emphasize locomotion, rather than sleep.

As noted above, we have rewritten the manuscript to focus on the sleep-regulatory role of *Neurocalcin*. Therefore, we have removed the locomotor activity plots previously described in Figure 1—figure supplements 2 and 4 in order to focus more appropriately on the sleep phenotype of *Neurocalcin* knockout and knockdown flies.

In Figure 1—figure supplement 2, activity counts per waking minute should be reported, not total beam breaks.

Using video-tracking velocity data from the DART system, we previously provided waking velocity data during the evening activity peak and the normally quiescent period during the night. We have also now included average velocity data across the entire 24 h period, showing reduced overall wake velocity in *Nca*^KO^ flies. These data are included in Figure 1—figure supplement 8A-D of the revised text.

[Editors' note: the author responses to the re-review follow.]

Essential Revisions:1) The story would benefit from a more mechanistic explanation, especially linking the data in Figure 3 to the data in Figure 6. One such experiment would be to manipulate the A05, C01, and OK107 knockdowns of neurocalcin (and relevant combinations) in the arousal assay, in DD, and in LL, similar to what was done in Figure 2 and Figure 3A and G for the pan-neuronal knockdown, and possibly combining the LL treatment with some of the functional imaging presented in Figure 5.

We thank the reviewers for these excellent suggestions. The resulting experiments (described in detail below, with data shown for the reviewer’s ease) have uncovered intriguing environment-specific effects of the two NCA-expressing neurons that we describe (termed *C01*- and *A05*- neurons), and demonstrated a role for light in regulating the excitability of neuropil regions within these domains.

Initially, we examined the effect of RNAi-mediated *Nca* knockdown in both *C01*- and *A05*- neurons in DD and LL. Similarly to pan-neuronal *Nca* knockdown (Figure 3A-B, G-H, manuscript), we found that *Nca* knockdown in *C01*- and *A05*- neurons in DD resulted in sleep loss during the subjective night, whereas sleep loss was suppressed in LL (Figure 4C-F). We also examined whether the nighttime arousal threshold was altered by *Nca* knockdown in *C01*- and *A05*- neurons. This was indeed the case, while daytime arousal was unaffected (Figure 4G-H). Thus, *Nca* knockdown in *C01*- and *A05*- neurons phenocopies the nighttime sleep loss and increased arousal observed in *Nca* mutants or following pan-neuronal *Nca* knockdown.

These data confirm the importance of NCA within *C01*- and *A05*- neurons for regulating night sleep/arousal, and are included in the main text in Figure 4C-H, described in the second paragraph of the subsection “NCA acts in two neuronal subpopulations to promote night sleep”.

Next, we reduced *Nca* expression in either *C01-, A05*- or *ok107*-neurons alone and assessed sleep in DD or LL. Surprisingly, *Nca* knockdown in *C01*-neurons did not impact sleep in DD but instead significantly reduced sleep in LL (Author response image 4). In striking contrast, *Nca* knockdown in both *A05*- and *ok107*-neurons reduced sleep in DD but not LL (Author response image 4).

**Author response image 4. respfig4:** Effect of *Nca* knockdown in *C01-, A05*- and *ok107*-neurons on sleep in LL and DD. (A-B) Mean sleep patterns following *Nca* knockdown in *C01*-neurons in constant dark (DD) (**A**) and constant-light (**B**) conditions. A: + > *kk*, n = 64, *C01* > +, n = 24, *C01* > *kk*, n = 24. B: + > *kk*, n = 76, *C01* > +, n = 51, *C01* > *kk*, n = 52.(C-D) Mean sleep patterns following *Nca* knockdown in *A05*-neurons in constant dark (DD) (**C**) and constant-light (**D**) conditions. C: + > *kk*, n = 64, *A05* > +, n = 51, *A05* > *kk*, n = 54. D: + > *kk*, n = 76, *A05* > +, n = 28, *A05* > *kk*, n = 32.(E-F) Mean sleep patterns following *Nca* knockdown in *ok107*-neurons in constant dark (DD) (**E**) and constant-light (**F**) conditions. E: + > *kk*, n = 64, *ok107* > +, n = 24, *ok107* > *kk*, n = 19. F: + > *kk*, n = 76, *ok107* > +, n = 24, *ok107* > *kk*, n = 21.(G-H) Median subjective night sleep (**G**) or total sleep (**H**) for the above genotypes in either constant-dark (**G**) or constant light (**H**). ns – p > 0.05, *p < 0.05, **p < 0.01, ***p < 0.001, compared to driver and RNAi alone controls, Kruskal-Wallis test with Dunn’s post-hoc test.

These experiments reveal a highly complex interaction between NCA levels in *C01-, A05*- and *ok107*-neurons and the external environmental condition, and suggest that the absence/presence of light, potentially in combination with circadian arrhythmicity in LL, increases or decreases the relative ability of *C01-, A05*- and *ok107*-neurons to promote wakefulness.

However, while these results are intriguing and will certainly form a platform for further interesting experiments, we feel that these data will likely confuse the readers of our paper, as we do not currently possess a robust understanding of how clock and light-sensing pathways differentially interact with each sub-circuit under LL or DD conditions.

Our manuscript will thus focus on simultaneous *Nca* knockdown in *C01*- and *A05*-neurons (or in one section, *ok107*- and *A05*-neurons) in either 8L: 16D or LL. In these genetic/environmental conditions, we possess sleep and optical imaging experiments that are well correlated (see below for synaptopHluorin data performed in LL) and which provide a mechanistic explanation for the suppression of *Nca*-knockdown-mediated sleep loss in LL.

In parallel to the above experiments we also examined whether *Nca* knockdown in *C01-, A05*- or *ok107*-neurons impacted arousal in 8L: 16D (Figure 6A-D). We found that *Nca* knockdown in either *C01*- or *ok107*-neurons significantly enhanced arousal during the night, resulting in the nighttime arousal threshold becoming more similar to the daytime arousal threshold (Figure 6A-D). Since *ok107*-Gal4 labels the mushroom body Kenyon cells (MB-KCs), and *C01*-Gal4 is expressed within the αβ-lobes of the MBs, these new data strongly suggest that NCA acts with MB αβ-KCs to regulate arousal during the night. This is exciting new data, and we have included these results in our manuscript in Figure 6, described in the subsection “NCA functions in the mushroom bodies to regulate sleep and arousal”.

We also examined whether NCA acts in *A05*-neurons to alter daytime or nighttime arousal (Author response image 5). Relative to controls, we found no significant difference in arousal following *Nca* knockdown in *A05*-neurons during the day or night (Author response image 5). However, we note that the *A05*-Gal4/+ control exhibits an unusually high degree of arousal during the night (Author response image 5). This has prevented us from drawing robust conclusions regarding the role of NCA in *A05*-neurons in regulating nighttime arousal, and thus in our manuscript we have focused on the role of NCA within the *C01/ok107*-domain in regulating nighttime arousal.

**Author response image 5. respfig5:** NCA acts in the mushroom bodies to regulate nocturnal arousal. (**A-B**) Percentage of adult male flies expressing *Nca* RNAi (*kk*) in *C01*-neurons (*C01* > *kk*) and control flies responding or not responding to vibration stimulus at either ZT4 (day; A) or ZT16 (night; B). ZT4: *C01* > +, n = 22, + > *kk*, n = 61, *C01* > *kk*, n = 27. ZT16: *C01* > +, n = 19, + > *kk*, n = 54, *C01* > *kk*, n = 21. (**C-D**) Percentage of adult male flies expressing *Nca* RNAi (*kk*) in MB-KCs (*ok107* > *kk*) and control flies responding or not responding to vibration stimulus at either ZT4 (day; C) or ZT16 (night; D). ZT4: *ok107* > +, n = 26, + > *kk*, n = 47, *ok107* > *kk*, n = 28. ZT16: *ok107* > +, n = 26, + > *kk*, n = 44, *ok107* > *kk*, n = 27. (**D-E**) Percentage of adult male flies expressing *Nca* RNAi (*kk*) in *A05*-neurons (*A05* > *kk*) and control flies responding or not responding to vibration stimulus at either ZT4 (day; A) or ZT16 (night; B). ZT4: *A05* > +, n = 19, + > *kk*, n = 61, *A05* > *kk*, n = 29. ZT16: *A05* > +, n = 20, + > *kk*, n = 54, *A05* > *kk*, n = 31. ns – p > 0.05, *p < 0.05, ***p < 0.001, Binomial test with Bonferonni correction for multiple comparisons.

The reviewers also requested us to examine how LL conditions impact neural excitability in *C01*- and *A05*-neurons following *Nca* knockdown. As shown in Figure 7A-B, *Nca* knockdown in *C01*- and *A05*- neurons in 8L: 16D enhanced neurotransmitter release from the MB αβ-KCs and neurons innervating the antennal mechanosensory motor center (AMMC). Since *Nca* knockdown in *C01*- and *A05*- neurons causes robust night sleep loss in 8L: 16D and DD but not in LL, we hypothesized that constant light might suppress this enhanced synaptic output. This was indeed the case (Figure 7E-H). In LL, synaptic output in the MB αβ lobes did not increase following *Nca* knockdown (Figure 7E), while in the AMMC synaptic output was actually reduced, not increased (Figure 7F). As in 8L: 16D, synaptopHlourin fluorescence was unchanged following *Nca* knockdown in LL in the MB γ-lobe region and the superior medial protocerebrum (SMP) (Figure 7G, H).

The data suggest that constant light suppresses the wake-promoting effect of *Nca* knockdown by blocking enhanced neurotransmitter release in the *C01/A05* circuit. Since the MB αβ-lobes and AMMC are components of the *C01*-Gal4 domain (Figure 5), and our new data shows that NCA acts in *C01*-neurons to regulate the arousal threshold during the night (Response Figure 3B, Manuscript Figure 6B), *C01*-positive domains such as the MB αβ-KCs and AMMC may be key regions that mediate the sleep-promoting effect of NCA.

These results are included in the main text in Figure 7E-H and described in the subsection “NCA inhibits synaptic output in a dark-dependent manner”.

2) While doing the experiments described in (1) would make the story more comprehensive, more detail may be necessary to link the findings related to cryptochrome to the neuronal populations described in Figure 6. Is neurocalcin's response to light also happening within the cryptochrome positive neurons, or are the responses of cryptochrome positive neurons interacting with the populations described in Figure 6 to regulate sleep?

We examined the above questions by knocking down *Nca* expression via transgenic RNAi in *cry*-positive neurons alone using the most widely expressed *cry*-Gal4 (*cry*-Gal4:16; Zao et al., 2003, Cell) alone or in combination with either *C01*- or *A05*-Gal4. We did not observe sleep loss in any of the above genotypes (*cry* > *kk, cry/C01* > *kk* or *cry/A05 > kk*, Author response image 6). This suggests that *cry*-positive neurons are not a sleep-relevant component of *C01/A05* neurons. One possibility, therefore, is that *cry*-neurons interact with *C01*- and/or *A05*-neurons to partially mediate the suppressive effect of light on sleep loss in *Nca* knockdown/mutant flies. However, given that the circuit logic of this effect remains unclear, we have not included these data in our manuscript.

**Author response image 6. respfig6:** NCA expression in *cry*-neurons is not required for *Nca^KD^*-mediated sleep loss. (A-B) Mean sleep pattern in 8L: 16D conditions (**A**) and median night sleep (**B**) for the indicated genotypes (see text). n= 20-51, ns – p > 0.05, Kruskal-Wallis test with Dunn’s post-hoc test.

3) The explanation for Figure 3 is somewhat confusing. Do control flies in the LL case exhibit similar quantities of sleep to neurocalcin knockdown because light inhibits neurocalcin function in the wild type case? This seems unlikely since in the presence of light, the controls do not show a reduction in sleep. The fact that the flies sleep more in constant light than in constant dark also seems extremely unusual.

We agree that light is unlikely to inhibit NCA function directly. Instead, our new data (Figure 7E-H) support a model in which light-sensing circuits intersect with *C01*- and/or *A05*-neurons and suppress their activity, which is normally enhanced by *Nca* knockdown to increase arousal and wakefulness.

To address the reviewers concern about constant light increasing sleep levels compared to constant dark, we first examined whether this could be due to a genetic background effect. We measured sleep levels in isogenic iso31 flies (our control stock) or in non-isogenic Canton-S male flies. In both cases, total sleep levels were significantly increased in LL compared to DD (Author response image 7). We performed identical experiments on males from driver- or transgene-containing stocks outcrossed into iso31 (*elav*-Gal4/+ or *Nca* RNAi/+) and found identical results (Author response image 7). Thus, in our recording conditions, constant light enhances sleep.

**Author response image 7. respfig7:** Constant light increases total sleep in adult male *Drosophila*. Median total sleep across 24 h in either constant-dark (DD) or constant-light (LL) are shown for adult males from four different genetic backgrounds. n = 15-64. *p < 0.05, ***p < 0.001, Kruskal-Wallis test with Dunn’s post-hoc test, with LL compared to both LD and DD.

While sleep levels in LL, LD and DD are rarely compared in the literature, other manuscripts support our observations. For example, in work performed by the Koh lab (Afonso et al., 2015; see Figure 2D and Supplementary figure 2), iso31 males and females do indeed appear to exhibit increased sleep in LL compared to DD. Other groups have also investigated sleep levels in constant light, for example, the Griffith lab (Parisky et al., 2016, Current Biology) and the Rosbash lab (Shang et al., 2008, PNAS). The baseline sleep levels under LL in these works do appear to be lower than our own. However, in both cases females rather than males were studied under LL. Given that male flies exhibit more daytime sleep compared to females (i.e. sleep during the light phase), this may contribute to the difference between our work and previous studies.

4) Which neurons express Nca protein, is it really expressed by all neurons as the authors indicate is previously shown? Teng et al., 1994, which is the only report of Nca expression in the fly, simply states that Nca is expressed "throughout the central nervous system" but only looked at adult brain slices with images that are too low quality to make any claim that all neurons express this protein. Furthermore, do Nca protein levels vary between sleep-wake states to explain why it is only important during sleep? Rat and likely other mammalian Hippocalcin antibodies are commercially available, given that the Teng et al. paper successfully used a rat antibody to label fly brains, it would be worth trying one or more of these antibodies both for immunohistochemistry to determine expression pattern and western blots to assess if Nca cycles circadian or in response to light/dark.

We agree that the NCA immuno-staining published in Teng et al., 1994, does not provide enough resolution to accurately define the precise expression pattern of NCA in the fly brain.

We undertook several independent approaches to attempt to fill this knowledge gap. We initially combined injection of two NCA peptide fragments (KIFRQMDRNKDGKLS and KMPEDESTPEKRTDK) in rabbits to raise a new NCA antibody (antibody service, Eurogentec inc). However, using these custom antibodies we were unable to detect NCA protein in the fly brain by western blot or immuno-staining.

Since the antibody generated by Teng et al. is raised against *Drosophila* full-length NCA (see Teng et al., 1994, Methods), we contacted the authors to request this antibody. However, we were unable to obtain this antibody. We also followed the reviewers’ suggestion to use an antibody against full-length human Hippocalcin (ab168214, Abcam plc UK), which is suitable for Immunohistochemistry. Unfortunately, we did not detect NCA-specific signals in the fly brain with this antibody.

Because of these technical constraints, we have been unable to characterise the expression pattern of NCA in the fly brain, and have therefore amended to text to state that NCA is ‘expressed in synaptic regions throughout the fly brain’ rather than ‘expressed in all neurons’.

We are in the process of generating knock-in flies with V5-tagged NCA under the control of the endogenous *Nca* promoter, which will enable the cellular and sub-cellular expression of NCA to be defined in detail. However, due to time constraints we are unable to add this data to the manuscript.

5) How does Nca inhibit neuronal activity in the context of sleep/wake or light/dark? The previous manuscript suggested it was dopamine receptors, now the Discussion mentions NMDA receptor internalization. Can this be tested genetically and/or by looking at protein localization?

To examine genetic interactions between NMDARs and *Nca*, we crossed a P-element insertion in *NMDAR1 (NMDAR1*^MI11796^) into the *Nca* knockdown background and tested for an epistatic interaction between the *NMDAR1* and *Nca* loci. Adding one copy of *NMDAR1*^MI11796^ mutation resulted in further reduction of night sleep in *Nca* knockdown flies (Author response image 8, red asterisks). However the same *NMADR1* mutation also caused significant sleep loss in the presence of the *elav*-Gal4 insertion (Author response image 8, blue asterisks). Hence, the enhancement of sleep loss in *Nca* knockdown flies appears to be additive. In our Discussion, we have clarified the text to make clear that NCA may be acting through multiple molecular pathways (potentially involving post-synaptic receptors and presynaptic ion channels) to regulate neurotransmitter release, similarly to its mammalian homologue Hippocalcin.

**Author response image 8. respfig8:** *Nca* and *NMDAR1* mutations additively affect sleep. (**A**) Mean sleep patterns of *Nca^KD^* males (*elav* > *kk*) and heterozygous controls with and without one copy of the *NMDAR1*^MI11796^ (*NMDAR1^MI^*/+) allele in 8L: 16D conditions. (**B**) Median night sleep levels in the above genotypes. Heterozygosity for *NMDAR1*^MI11796^ resulted in sleep loss in the background of *elav*-Gal4/+ controls, and reduced sleep further in an additive manner in *elav* > *kk* males. *elav* > *kk; NMDAR1*^MI^/+: n = 32; *elav* > +; *NMDAR1*^MI^/+: n = 26; + > *kk; NMDAR1*^MI^/+n = 38; *elav* > *kk*: n = 32; *elav* > +: n = 30; + > *kk*: n = 34. ns – p > 0.05, *p < 0.05, **p < 0.01, ***p <0.001, Kruskal-Wallis test with Dunn’s post-hoc test.

6) The Results and Discussion points about the distinct roles of A05 and C01 (pro-arousal and modulatory, respectively) feel over-interpreted.

We agree with the reviewers that the assignment of *A05*- and *C01*-neurons as either pro-arousal or modulatory is over-simplified. In 8L: 16D, thermogenetic excitation of these neurons using TrpA1 (Figure 8) suggests that *C01*-neurons are strongly wake-promoting, whereas activation of *A05*-neurons only promotes wakefulness in the context of parallel *C01*-neuron activation. However, the reviewer’s suggestions have led us to find that both neuronal subsets appear capable of promoting wakefulness following *Nca* knockdown depending on the environmental condition (LD, DD or LL). Thus, in the Discussion we have avoided referring to *A05*-neurons as ‘modulatory’. Because of the complexity of interpreting the above DD/LL data, in this paper we will focus specifically on 8L: 16D and LL, and throughout the manuscript we are careful to state which environmental condition our results pertain to.

[Editors' note: further revisions were requested prior to acceptance, as described below.]

1) Figure 4H: was the axis label supposed to read "percentage of flies responding to night time stimulus" (rather than daytime stimulus)?

We thank the reviewer for highlighting this error, which we have now corrected.

2) There was a typo in an earlier review when it was requested the authors report the combinations of Gal4 drivers that did not yield negative results. This should have asked what combinations of Gal4 drivers the authors did test, since this kind of negative data is useful for interpreting how exclusive the network of neurons involved is. As it stands, the reason why the C01 and A05 Gal4 drivers were chosen as the specific combinations seems a little odd.

We have added negative data from eight additional combinations of driver lines to Figure 4—figure supplement 1B.

We chose these combinations, as well as the *C01/A05* driver combination, in a hypothesis-based manner. Firstly, we posited that dopaminergic circuits (either dopamine-releasing (*ple*-Gal4) or Dop1R1-expressing (*C01*-Gal4 and *R72B07*-Gal4) neurons) might be involved in the regulation of sleep by NCA, since these circuits are known to influence sleep/wakefulness in Drosophila (Kume et al., (2005); Ueno et al., (2012); Liu et al., (2012)). Secondly, since light-sensing pathways modulates the wakefulness in *Nca* knockdown flies, we investigated two circuits that transmit light information: tubercular-bulbar neurons, subsets of which we and others have demonstrated are sleep promoting (Lamaze et al., (2018); Guo et al., (2018)), and cry-expressing neurons. Tubercular-bulbar neurons are labelled by the *A05*-Gal4 driver and additional drivers shown in Figure 4—figure supplement 1B.

Of these combinations, only expression of *Nca* RNAi in *C01/A05*-neurons reduced night sleep, suggesting that NCA expression within specific neuronal components of the *C01*- and *A05*-Gal4 domains regulate night sleep.

3) Results section, fifth paragraph: "To test whether sleep loss caused by neuronal Nca knockdown flies was due to [...]"The authors ask if the clock is affected by NCA KD and show constant dark records. Under what light conditions were those flies entrained? It would be useful to also show the activity patterns under LD conditions, preceding release into DD.

We have modified Figure 1—figure supplement 5 to more clearly show the first day in LD preceding release into DD, and have included details of how these flies were entrained in the Materials and methods section. Briefly, flies were entrained in 12L: 12D cycles for three days before transfer to DAM tubes, after which fly activity was recorded for one day in 12L: 12D before release to DD.

4) The authors test a model of NCA action in A05 and C01 neurons in two ways. First with TrpA1 activation. Second with dORK silencing. I find the first experiment reasonable, but the second experiment much less compelling. The model postulates that A05/C01 neuron activity is regulated by NCA and that more activity by those neurons promotes waking and disrupts night time sleep. However, constitutive dORK expression is known to irreversibly damage neurons. The dORK over-expression experiment shows that the loss of sleep in the Nca KD requires A05/C01 neurons, but does not distinguish between suppression of neuronal activity, or a non-specific decline in neuron health. As I understand it, either outcome could produce the observed result.

While we appreciate the reviewer’s concerns regarding the potential of *dORK*-∆C2 to damage neurons, we have been unable to find evidence in the prior literature for an irreversible damage of neurons by expression of this transgene. In contrast, multiple lines of evidence indicate that *dORK*-∆C2 does not alter neuronal development or formation of synaptic contacts. For example, Nitabach et al., (2002) showed that expression of *dORK*-∆C2 in PDF-expressing clock neurons (the s-LN_v_s) does not alter their development or axonal path-finding. Similarly, Kremer et al., (2010) showed that expression of *dORK*-∆C2 in olfactory projection neurons suppresses action potential firing but does not impact expression of the active zone protein Bruchpilot or prevent their proper synaptic innervation of the mushroom body calyx region. We further note that *dORK*-∆C2 has previously been used to study sleep circuitry in Drosophila (for example, Tabuchi et al., (2015)).

Nonetheless, we have attempted to allay the reviewer’s concerns by investigating whether the phenotypes associated with silencing of *C01*- and *A05*-neurons with *dORK*-∆C2 are consistent with ‘irreversible damage’ to these cells.

Firstly, as a positive control to assess the impact of damaging *C01*- and *A05*- neurons, we expressed the pro-apoptotic genes *hid* and *reaper* in either *C01*- or *A05*-neurons alone. Inducing cell death in either population resulted in 100% lethality prior to the adult stage. In stark contrast, expression of *dORK*-∆C2 in both *C01*- and *A05*-neurons simultaneously had no apparent impact on adult-stage viability.

Secondly, we complemented the above experiment by re-analysing our sleep data to determine whether expression of *dORK*-∆C2 in *C01*- and *A05*-neurons impacted adult locomotor patterns. Since ablation of *C01*- or *A05*-neurons is lethal, if *dORK*-∆C2 indeed resulted in partial yet irreversible damage, we would likely observe an effect on locomotor patterns despite the viability of these flies. Author response image 9 shows activity patterns of otherwise wildtype iso31 flies expressing *dORK*-∆C2 in *C01*- and *A05*-neurons compared to driver and transgene alone controls. Expression of *dORK*-∆C2 in *C01*- and *A05*-neurons did not impact overall activity levels (Author response image 9) or peak activity levels at ZT8 (Author response image 9). These results are inconsistent with the postulate of *dORK*-∆C2 causing irreversible damage to *C01*- and *A05*-neurons.

Thirdly, we complemented our experiments with constitutive *dORK*-∆C2 expression by expressing a temperature-sensitive inhibitor of synaptic vesicle endocytosis (*shi*[ts]) in *C01*- and *A05*-neurons alongside the kk *Nca* RNAi construct. By shifting flies to 31ºC we can suppress neurotransmitter release from *C01*- and *A05*-neurons in a *Nca* knockdown background. Our *dORK*-∆C2 results predict that this manipulation should inhibit sleep loss due to *Nca* knockdown. However, if this phenotype was solely due to a non-specific decline in neuronal health rather than an inhibition of synaptic release, we would expect acute silencing of *C01*- and *A05*-neurons to have no impact on sleep levels following *Nca* knockdown.

As shown in Author response image 9, compared to control flies expressing an FRT-stop-FRT-GFP transgene, expression of *shi*[ts] alongside *kk Nca* RNAi significantly increases sleep levels at 31ºC, suggesting that *Nca* is required acutely to regulate sleep in *C01*- and *A05*-neurons.

We note that night sleep in *Drosophila* is strongly reduced at 31ºC (Lamaze et al., (2017)), likely limiting the rescuing effect *shi*[ts]. Therefore, we would ideally repeat these experiments at a slightly lower temperature of 29ºC, alongside controls expressing *shi*[ts] in *C01*- and *A05*-neurons in the absence of *kk Nca* RNAi. Therefore, we have not included this data in the manuscript.

Collectively, the above data support the premise that constitutive *dORK*-∆C2 expression in *C01*- and *A05*-neurons does not result in cell death/damage during the time-window in which we recorded sleep (early adulthood). Rather, *dORK*-∆C2 suppresses neurotransmitter release (similarly to UAS-*shi*[ts] but constitutively rather than acutely), and this counteracts the enhanced neurotransmitter release caused by *Nca* knockdown that normally results in enhanced wakefulness during the night.

**Author response image 9. respfig9:** Acute silencing of C01- and A05-neurons suppresses sleep loss induced by Nca knockdown. (**A**) Activity in adult male flies expressing UAS-*dORK*-∆C2 in *C01/A05*-neurons, as quantified by infrared beam breaks using the DAM system. Driver and transgene alone controls are also shown. (**B**) Box plots comparing median peak activity at ZT8 (the evening activity peak). No significant difference in peak activity was found following constitutive silencing of *C01/A05*-neurons. *C01/A05* > +, n = 38; + > *dORK*-∆C2, n = 26, *C01/A05* > *dORK*-∆C2, n = 34. ns: p>0.05, Kruskal-Wallis test with Dunn’s post-hoc test. (C-D) Average sleep patterns (**C**) and total night sleep levels (**D**) in 8L: 16D at 31ºC of adult males expressing *kk Nca* RNAi in *C01/A05*-neurons alongside the acute inhibitor of synaptic vesicle endocytosis (*shi*[ts]) or a control transgene (FRT-stop-FRT-GFP). Note the significant increase in night sleep upon inhibiting synaptic release from *C01/A05*-neurons with reduced NCA levels (**D**). *C01/A05* > *kk*, FRT-stop-FRT-GFP, n = 25; *C01/A05* > *kk, shi*[ts] n = 22. *p

5) "Instead, sleep-relevant NCA activity can largely be localized to two distinct domains of the Drosophila nervous system defined by the A05- and C01-Gal4 drivers"This conclusion is based on spatially limiting the RNAi effects, thus defining brain regions that are necessary. It does not preclude involvement of other regions. Therefore this conclusion over-interprets the data.

We have modified the Discussion to state that “sleep-relevant NCA activity is necessary within two distinct domains of the Drosophila nervous system defined by the *A05*- and *C01*-Gal4 drivers”, rather than “largely localized to”.

6) Figure 1—figure supplements 2 & 6: the cartoon representations of Nca and its immediate neighbor CG7646 are confusing. In supplement 2, Nca and CG7646 appear to share two 5'UT exons in common. In supplement 6, CG7646 appears to be distinct and non-overlapping. Which is true? Did the authors test the sleep/activity behaviors of flies with CG7646 KD by RNAi?

To clarify, Figure 1—figure supplement 2A shows the mRNA transcript isoforms of *Nca* and its neighbour, cg7646, since the *kk, hmj* and *jf* dsRNAs target *Nca* mRNA rather than the *Nca* genomic DNA. *Nca* and *cg7646* do indeed share common upstream 5’ UTR elements. These are shown in Figure 1—figure supplement 2A, where each *Nca* and *cg7646* mRNA isoform is shown. We have modified this figure to remove any irrelevant non-transcribed regions from each of the *Nca* and *cg7646* isoforms shown in Figure 1—figure supplement 2A. We also modified the corresponding figure legend to emphasise that transcript isoforms are shown in this figure supplement.

In Figure 1—figure supplement 6A we depict the genomic region (and surrounding sequences) corresponding to the targeting arms used for homologous recombination. The two common 5’ UTR elements depicted in Figure 1—figure supplement 2A are not contained within the targeting arm sequences and are therefore not shown in Figure 1—figure supplement 6A.

Finally, we did test whether knockdown of *cg7646* impacted sleep using two NIG RNAi lines. As shown in Author response image 10, in contrast to *Nca* knockdown, neither RNAi line targeting *cg7646* altered total night sleep levels.

**Author response image 10. respfig10:** Median night sleep levels following pan-neuronal expression of two independent RNAi lines targeting *cg7646* mRNA compared to driver (*elav*Gal4/+) and RNAi transgene alone controls. No significant difference in night sleep following *cg7646* knockdown using either RNAi construct. n = 12-15. Ns: p>0.05, Kruskal-Wallis test with Dunn’s post-hoc test.

[Editors' note: further revisions were requested prior to acceptance, as described below.]

[...] I suggest the important issue is whether prolonged hyperpolarization of Drosophila neurons is (beyond reasonable doubt) an effective means to exclusively affect neurotransmission. Ceriani and colleagues in 2011 showed that prolonged expression of Kir in LNv pacemaker neurons irreversibly affected molecular oscillations, structural maturation and neuropeptide content. In contrast, restricting Kir expression to adult stages, led to outcomes that were at least partially reversible. Therefore, I hold to the opinion that the present dORK experiment is not compelling, in spite of the additional observations provided. If the authors insist on its inclusion, then I think it's important to share with readers a clear caveat based on the Ceriani work.

We respect the reviewer’s concerns regarding potential detrimental effects of constitutive *dORK*-∆C2 and have endeavoured to modify the text accordingly to include such a caveat.

Regarding the work by Ceriani and colleagues (Depetris-Chauvin et al., 2011), the authors of this manuscript show that prolonged (but not acute) Kir2.1 expression impacts molecular clock oscillations in PDF neurons. Yet importantly, they also show that action potential firing in adult PDF neurons constitutively expressing Kir2.1 can be acutely restored to wild-type levels by application of 200 µM Ba2+, a Kir2.1 channel blocker (see Figure 2B, Depetris-Chauvin et al., 2011). From this, the authors conclude the following: “Therefore, these results demonstrate that both transient and persistent expression of kir2.1 reliably abolish firing of PDF neurons *without affecting their viability*” (our italics). Given this conclusion, we hope the reviewer understands that we are reticent to cite this manuscript as evidence that prolonged Kir2.1 expression impacts neuronal viability.

However, the same authors indeed later postulate that prolonged Kir2.1 expression may “ultimately impinge on cell viability” (Discussion). Yet the data shown in this manuscript solely pertains to the function of the circadian oscillator, and there is no evidence presented of an effect of Kir2.1 on noncircadian cellular processes critical for neuronal viability.

Combined with data from Nitabach et al., (2012) (see Figure 2), the current literature suggests that prolonged Kir2.1 expression in PDF neurons does not impact either their ability to fire action potentials nor their development or axon guidance. We also note that other work from the Nitabach lab (Figure 1, Wu et al., (2008)) indicates that *dORK*-∆C2 reduces the resting membrane potential of PDF neurons by ~ 16 mV while Kir2.1 reduces RMP by ~ 27 mV. Thus, the *dORK*-∆C2 transgene that we use has a subtler effect on neuronal excitability compared to Kir2.1.

Nonetheless, the reviewer is correct that there is evidence in the literature for a negative impact of constitutive expression of specific ion channels on the viability of certain neuronal subtypes. For example, Nadeau et al., (2000) showed that Kir1.1 expression induces apoptosis of mammalian hippocampal neurons. We have therefore inserted the following text in the Discussion that cites both Nadeau et al., and Depetris-Chauvin et al., 2011 and includes caveats regarding constitutive ion channel expression. We hope the reviewer finds this alteration acceptable.

“Ex vivo imaging demonstrates that *Nca* knockdown enhances synaptic output from subsets of *C01*- and *A05*-neurons innervating the MB αβ-lobes and the AMMC. [...] Yet when NCA expression is inhibited in *C01*- and *A05*-neurons simultaneously, the resulting enhancement of synaptic output within this wider network is sufficient to reduce night sleep.”

Consideration of CG7646 - I appreciate the clarification of the complicated genomic arrangement of CG7646 and Nca. It is helpful to know they share 5'UTR sequences but different CDS exons. I think it would be useful to make this point clear in the text - for example on line 111 of the revision, CG7646 is described as a "neighboring" locus. I interpret "neighbor" as independent or free-standing. These two genes are overlapping or in some fashion "compound". I recommend the text include a clear description of this genomic arrangement and be re-examined to ensure clarity on this point in all mentions. Likewise, I suggest inclusion of the RNAi knockdown results of CG7646 as supplemental information to allay concerns on this point. Regarding that experiment, one other minor point: in the response and author response images, the authors describe using two different RNAi's from NIG. However, my reading of the NIG site indicates availability of only one RNAi for CG7646, albeit offered as a II and III chromosome insertion. The figure leads the reader to think two different RNAis were tested. The text should be clear as to whether the same RNAi at different insertion sites, or different RNAi's were tested.

We have altered the text to emphasise that *cg7646* and *Nca* share common 5’ regulatory elements and removed any description of the two loci as ‘neighboring’. We have included data showing the effect of *cg7646* RNAi on sleep in Figure 1—figure supplement 2K. In the figure legend of Figure 1—figure supplement 2K we have stated that two different chromosomal insertions of the same RNAi hairpin were used to knockdown cg7646. These lines are also described in the Key Resources Table and the Materials and methods sections.